# NAC couples protein synthesis with nascent polypeptide myristoylation on the ribosome

Sara Zdancewicz ⬤[1], Emir Maldosevic ⬤[1], Kinga Malezyna[1] & Ahmad Jomaa ⬤[1,2✉]

## Abstract

**N-glycine myristoylation allows for reversible association of newly synthesized proteins with membranes to regulate essential functions such as cellular signaling and stress responses. This process can be catalyzed during protein synthesis by N-myristoyltransferases (NMTs), and its dysregulation has been implicated both in cancer and heart disease. Although the nascent polypeptide-associated complex (NAC) orchestrates the binding of several co-translational processing factors on ribosomes, its role in facilitating nascent protein myristoylation by NMT2 remains unclear. Here, we show that NAC mediates binding of NMT2 to translating ribosomes, which together form an extended channel that guides the nascent chain as it emerges from the polypeptide exit tunnel to the catalytic site of NMT2. Furthermore, the ternary ribosome:NMT2:NAC complex is stabilized by a ribosomal RNA clamp that, together with NAC, orients NMT2 on the ribosomal surface for co-translational myristoylation of nascent chains. Our work uncovers the molecular mechanism coupling protein synthesis to nascent protein myristoylation and underscores the role of NAC as a master regulator of protein biogenesis on the ribosome.**

**Keywords** NMT2; Translating Ribosome; NAC; N-myristoyltransferases; Nascent Chain; Cryo-EM
**Subject Categories** Structural Biology; Translation & Protein Quality

## Introduction

In eukaryotes, protein modifications such as glycosylation, lipidation, and ubiquitylation play critical roles in determining protein maturation, folding, and subcellular localization (Morales-Polanco et al, 2022; Gamerdinger and Deuerling, 2024; Chang, 2023). These modifications regulate protein function to modulate essential cellular processes involved in cell signaling, survival, and immune response (Hu et al, 2010; Bouamr et al, 2003; Adam et al, 2007). One of these essential modifications is N-glycine myristoylation and is characterized by the addition of a 14-carbon fatty acid to the N-terminus of a nascent chain polypeptide (Wang et al, 2021; Wilcox et al, 1987). Myristoylation is carried out by a family of enzymes known as N-myristoyltransferases (NMTs), which are cytosolic enzymes that have two GNAT-like

(Gcn5-related N-acetyltransferase) domains (Dian et al, 2020). There are two isoforms of NMT—NMT1 (Duronio et al, 1992) and NMT2 (Giang and Cravatt, 1998)—with tissue-specific expression and overlapping but distinct substrates (Ducker et al, 2005). An important result of nascent chain myristoylation is that, once folded and functional, the protein is able to reversibly interact with membranes (Peitzsch and McLaughlin, 1993; McLaughlin and Aderem, 1995), making the myristoyl group a key component of proper protein localization in a variety of cellular processes (Resh, 1989; Franco et al, 1995; Randazzo et al, 1995). Dysregulation of myristoylation can severely impact human health, manifesting in various diseases ranging from brain or colorectal cancer to cardiac hypertrophy (Wright et al, 2010; Yuan et al, 2024; Zhang et al, 2022; Tomita et al, 2023). Although NMT1 and NMT2 share ~77% sequence identity (Giang and Cravatt, 1998), NMT2 in particular plays a significant role in cardiac remodeling and heart failure, with affected patients exhibiting up to a 60% decrease in cardiomyocyte NMT2 levels compared to healthy individuals (Tomita et al, 2023). It also remains unclear how NMT2 selects for substrates in the cell.

Previous biochemical and structural work elucidated how the myristoylation of substrates occurs and defined the catalytic site of the enzyme at the molecular level (Dian et al, 2020; Rivière et al, 2022; Castrec et al, 2018). In particular, NMT2 has a strong preference for an exposed glycine residue in a GXXXSK motif on the N-terminus of the protein to be modified (Castrec et al, 2018; Farazi et al, 2001b). Consequently, N-glycine myristoylation is a co-translational process that takes place as the nascent chain is still being synthesized. Localization to the ribosome has previously been shown by isolation of subcellular ribosomal fractions which contain high levels of NMT compared to elsewhere within the cell (Glover et al, 1997; Takamune et al, 2010).

Before NMT2 can modify substrates, methionine aminopeptidase (MAP) first cleaves off the initiator methionine as the nascent chain emerges from the polypeptide exit tunnel.(Gamerdinger and Deuerling, 2024; Varland et al, 2015; Giglione et al, 2015) This exposes the glycine residue to be modified and allows NMT2, which carries a donor molecule of myristoyl-CoA, to bind the substrate polypeptide and transfer the myristoyl group. Due to the stepwise nature of this substrate modification, N-glycine myristoylation is likely to be a spatio-temporally coordinated process, but the regulation of enzyme recruitment to ribosomes carrying myristoylation targets is not yet fully understood. Additionally, it is unclear whether NMT2 requires direct docking on the ribosome surface to modify newly synthesized nascent chains.

[1]Department of Molecular Physiology and Biological Physics, University of Virginia, Charlottesville, VA, USA. [2]Department of Biochemistry and Molecular Genetics, University of Virginia, Charlottesville, VA, USA. ✉E-mail: ahmadjomaa@virginia.edu

 

The nascent polypeptide-associated complex (NAC) is a ribosome anchored sorting factor that controls the access of enzymes and targeting factors to the ribosome depending on the identity of the nascent chain emerging from the polypeptide exit tunnel (Nyathi and Pool, 2015; del Alamo et al, 2011; Wiedmann et al, 1994; Raue et al, 2007). NAC is a ubiquitously expressed heterodimer composed of NACα and NACβ subunits and is conserved in all eukaryotes. The subunits dimerize to form a barrel domain (Spreter et al, 2005; Liu et al, 2010), a ribosome anchor (Wegrzyn et al, 2006), and three long, flexible tails that recruit different enzymes and targeting factors to the ribosome. In particular, the NACβ C-terminal tail recruits MAP1 (Gamerdinger et al, 2023), while the NACα N-terminal UBA domain recruits either N-acetyltransferase A/E or the signal recognition particle (SRP) to the ribosome (Jomaa et al, 2022; Klein et al, 2024; Lentzsch et al, 2024). However, whether NAC can also recruit NMT2 to the ribosome to facilitate substrate processing remains unknown.

Here, we show that human NMT2 binds to ribosomes, and its binding is enhanced by NAC. To further understand the molecular interplay of these factors, we then determined the structure of the ternary complex formed of NMT2, NAC, and the translating ribosome. The structure reveals a nascent chain substrate of NMT2 positioned in the catalytic site after it emerges from the polypeptide exit tunnel. We also resolve extensive contacts between NAC, NMT2, and ribosomal proteins and RNA, including an rRNA clamp by helix 59 to anchor NMT2 on the ribosome for efficient substrate engagement. Disrupting these interactions impaired NMT2 binding to the ribosome. Furthermore, we show that NAC recruits NMT2 to the ribosome using the C-terminal tail of NACβ. Together, our findings uncover the initial stages of N-glycine myristoylation on the ribosome by NMT2, highlighting NAC's pivotal role in orchestrating this essential modification in humans.

## Results

### NAC enhances NMT2 binding to translating ribosomes

Previously, N-myristoylation activity was shown to be associated with the ribosomal subcellular fractions of human lymphoblastic leukemia and cervical carcinoma cells (Glover et al, 1997). However, it is not clear whether NMT2 binds directly to the ribosome to myristoylate its target nascent chains. To understand NMT2 and ribosome interaction and the potential interplay with NAC (Fig. 1A), we first confirmed NMT2 localization in ribosomal fractions from human embryonic kidney cells (HEK293) using sucrose gradient centrifugation. NMT2 was detected in all ribosomal fractions from HEK293 cells (Fig. 1B). We also analyzed NAC migration and confirmed its presence within the ribosomal fractions as previously shown (Raue et al, 2007). Notably, both NMT2 and NAC signals were stronger in the 80S fractions than in the polysome fractions. We then analyzed NMT2 interactions with purified human ribosomes using an in vitro binding assay both in the presence and the absence of NAC. Our in vitro binding experiments indicated that NMT2 stably binds to ribosomes, with a two-fold increase in NMT2 binding observed in the presence of NAC (Fig. 1C,D; Appendix Fig. S1A).

Next, we tested whether NMT2 binding to translating ribosomes was impacted by the display of different nascent chain lengths of an

NMT2 substrate. To investigate this, we used an in vitro translation system from rabbit reticulocyte lysate (RRL) to synthesize myristoylated alanine-rich C-kinase substrate (MARCKS), a substrate specific to NMT2 (Tomita et al, 2023). The N-terminal myristoyl group regulates the ability of MARCKS to localize to the plasma membrane, and is important in a variety of cellular processes ranging from actin cytoskeleton modulation to $PIP_2$ sequestration to regulate cell motility (Aderem, 1992; Sheats et al, 2015; Chen et al, 2021). To expose the N-terminal glycine residue, we engineered a 3C cleavage site into the sequence to remove the initiator methionine (Fig. 1E, Appendix Fig. S1B). We designed 5 constructs with different nascent chain lengths varying between 35 and 200 amino acids in length. The N-terminal glycine residue will be in the tunnel in the shortest nascent chain (35 amino acids) and fully exposed in the 55–200 amino acid constructs. Isolated ribosome-nascent chain complexes ($RNC_{MARCKS}$) were then reacted with NMT2 and NAC to assess binding. Our results indicate that NMT2 interacts with $RNC_{MARCKS}$ and empty ribosome similarly with a modest increase to $RNC_{MARCKS}$ displaying 74 amino acids (Fig. 1F). Combined, our results show that NMT2 binds to translating ribosomes and this binding is further enhanced by the addition of NAC.

### NAC coordinates NMT2 binding and substrate engagement on the ribosome

To elucidate how NMT2 engages the substrate on $RNC_{MARCKS}$ and its interplay with NAC, we determined the structure of the ternary $RNC_{MARCKS}$:NMT2:NAC complex using single-particle cryo-electron microscopy (cryo-EM) (Fig. 2A). For the cryo-EM analysis, NAC and NMT2 were reacted with $RNC_{MARCKS}$ containing a 74 amino acid long nascent chain to maximize the display of N-glycine outside the polypeptide exit tunnel. Similar lengths were shown to be optimal for binding and processing by other co-translational factors (Lentzsch et al, 2024; Jomaa et al, 2021). To then trap NMT2 in an active, substrate-engaged state, myristoyl coenzyme A (myristoyl-CoA) was added to the reaction right before sample freezing for cryo-EM data collection and processing (see "Methods").

Initial 2D and 3D refinements resolved a strong density for both NMT2 and NAC on the ribosome (Fig. EV1A). Further focused 3D classifications on the enzymes improved the local density of NAC and resolved the substrate-engaged NMT2 (Fig. EV1B). The ternary complex was determined at an overall resolution of 2.8 Å (Fig. EV1C). Several regions of NAC and NMT2 were observed at side chain resolution (Fig. EV2A–F), including densities for the nascent chain both in the polypeptide exit tunnel of the ribosome and the catalytic core of NMT2 (Figs. 2B–E and 3A).

The cryo-EM density for NMT2 was resolved near the polypeptide exit tunnel, where individual secondary structure elements and the side chains of some alpha helices contacting the ribosome could be distinguished. This allowed for fitting the crystal structure of NMT2 as a rigid body with minimal changes (Figs. 2C and EV3A,B). In the structure, NMT2 makes contact with ribosomal RNA and protein, and with the NAC barrel domain, which is docked at a similar location as observed in previous ribosome structures (Fig. EV4A–D) (Lentzsch et al, 2024).

The obtained structure reveals that the catalytic site of NMT2 is positioned directly above the polypeptide exit tunnel. Together with

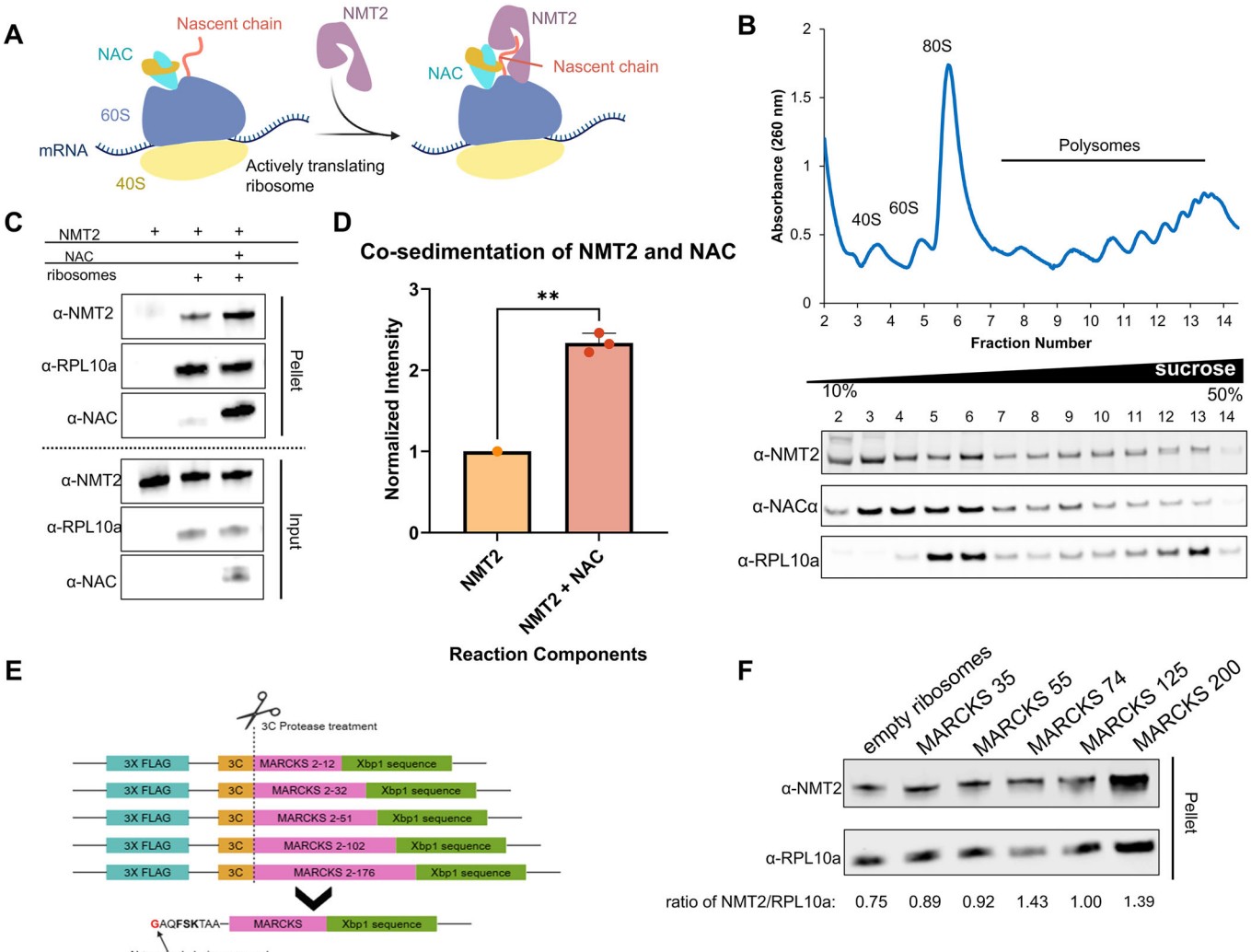

**Figure 1. NMT2 interaction with ribosomes in human cells, with NAC, and nascent chains.**

(A) Schematic of hypothesized mechanism of NMT2 recruitment and binding to the ribosome. (B) Profile of ribosome and polysome distribution through a 10–50% sucrose gradient of HEK293 cell lysate measured by absorbance at 260 nm. NMT2 and NAC distribution in fractions were analyzed by immunoblot, and the presence of ribosomes was confirmed by blotting for RPL10a. (C) Immunoblot analysis of co-sedimentation of NMT2 in the presence and absence of NAC. (D) Quantification of NMT2/NAC co-sedimentation assay. Three independent replicates of the assay were completed, and band intensity of the immunoblot was quantified in ImageJ as described in Methods, weighting NMT2 band intensity by RPL10a load per lane. Significance was determined at $p = 0.0026$ using a two-tailed paired $t$ test. Bar graph depicts the mean of three replicates with error bars showing standard deviation (**$p < 0.01$). (E) Schematic depicting the generation of different-length nascent chain constructs. All constructs contained an Xbp1 stalling sequence and a 3X FLAG tag for purification. A 3C cleavage site was engineered to expose an N-terminal glycine once treated. (F) Immunoblot showing NMT2 binding to RNC with varying length MARCKS constructs purified from RRL and NMT2 binding to empty ribosomes purified from HEK293 cells as a comparison. The ratio of NMT2 intensity to RPL10a intensity is listed below each lane. Source data are available online for this figure.

the NAC barrel domain, NMT2 binding to the ribosome forms a channel extending the peptide exit tunnel by ~44 Å (Fig. 3B). The newly formed channel may act to guide the emerging nascent polypeptide to the catalytic site of NMT2 as it exits the ribosome (Fig. 3C). Remarkably, the first nine amino acids of MARCKS were resolved in the catalytic core of NMT2, with visible side-chain densities and a density for Coenzyme A (CoA) in the active site (Figs. 2D and EV5A–D). Although the glycine residue is resolved within the catalytic site, no density for the myristoyl group (14-C) is visible, suggesting it may become flexible after being transferred to the nascent chain. EM densities for the side chains of nascent chain residues S6 and K7, as well as for some NMT2 residues lining

the nascent chain binding pocket, are visible in the structure. This enabled assessment of the interactions that stabilize the nascent chain within the NMT2 catalytic site (Figs. 2D,E and EV5D,E). Notably, S6 is positioned within range to form an electrostatic interaction with H300 of NMT2. K7 fits into a surrounding negatively charged pocket formed by D185, D186, D187, and D473, and in our structure, is within distance to form a salt bridge with D185.

Furthermore, the catalytic site of NMT2 is gated by loops rich in negatively charged and polar residues (Fig. EV6A,B,D). Compared to the previously reported apo yeast NMT structure (Farazi et al, 2001a) (Fig. EV6C), these loops undergo a conformational change

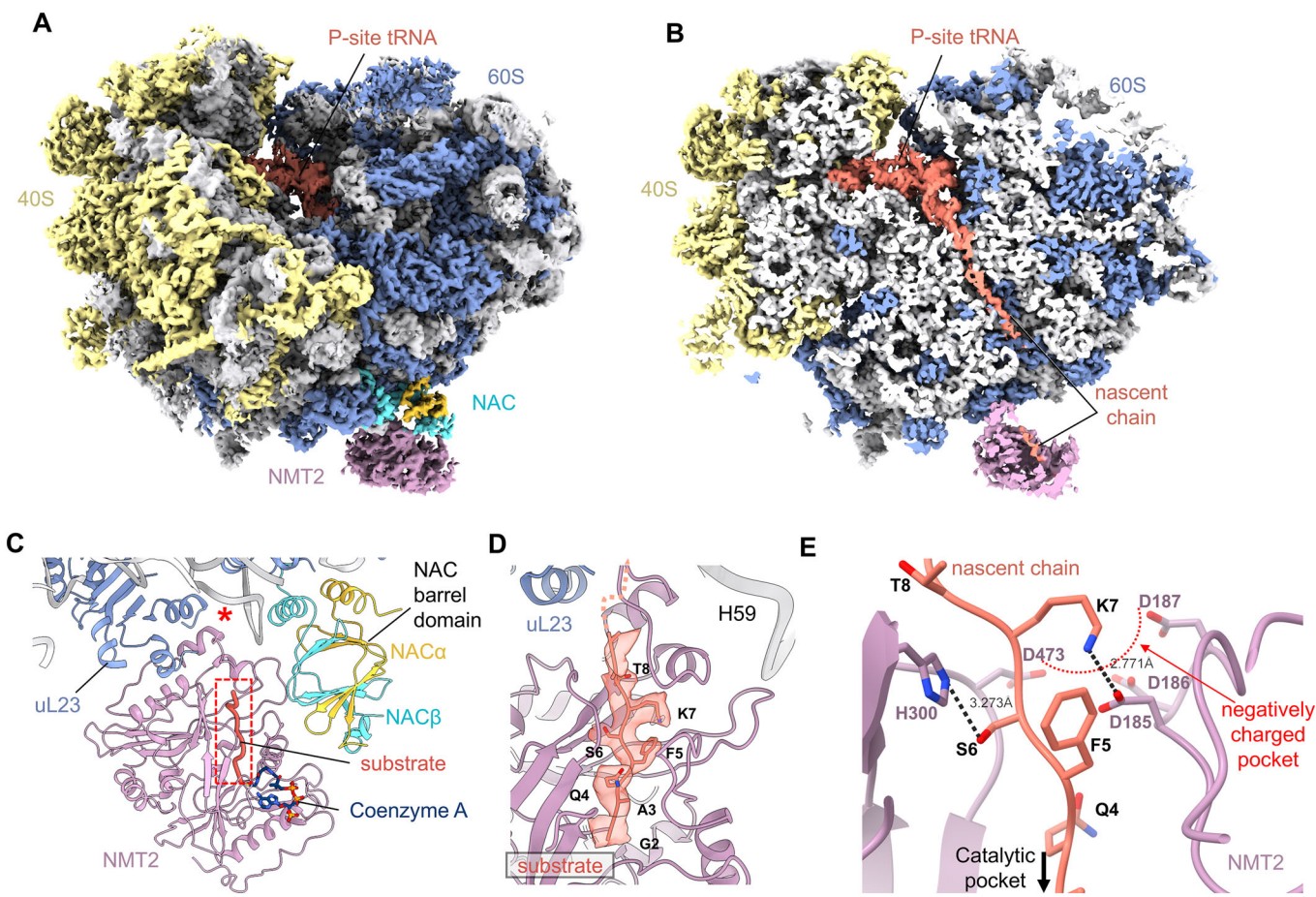

**Figure 2. Cryo-EM structure of the ternary RNC_MARCKS:NMT2:NAC complex.**

(**A**) Cryo-EM map of the ternary complex showing both the large ribosomal subunit (proteins in blue, rRNA in gray) and small subunit (proteins in yellow, rRNA in gray) as well as density for NMT2 (pink) and NAC (cyan, gold), also containing p-site tRNA density (coral). (**B**) A slab through the full ternary complex depicting the full P-site tRNA with attached nascent chain in the polypeptide exit tunnel (coral). Cryo-EM density for the nascent chain is also present and visible within the catalytic core of NMT2 (also shown in coral). (**C**) A close-up of the atomic model of NMT2 and NAC bound to the ribosome with the substrate nascent chain (coral) depicted as cartoons, as well as Coenzyme A (navy) shown as a stick model. The location of the polypeptide exit tunnel is indicated with a red asterisk. (**D**) Cryo-EM density corresponding to the nascent chain is shown with visible side chains for the first 9 amino acids of MARCKS engaged in the catalytic site of NMT2. (**E**) A close-up of electrostatic interactions (shown as black dashed lines) between residues 6 and 7 of the substrate nascent chain (coral) and NMT2 sidechains (pink). A negatively charged pocket formed by NMT2 residues 185, 186, 187, and 473 is indicated with a red dashed line.

upon substrate binding, sealing part of the NMT2 catalytic site (Fig. EV6E,F). This closed-loop conformation is similar to that observed in the crystal structure of human NMT1 bound to a peptide substrate (Dian et al, 2020). Taken together, the structure of the ternary complex visualizes an active state of NMT2 with the nascent chain trapped in the catalytic site during its modification on the translating ribosome.

## H59 rRNA clamps NMT2 for efficient substrate engagement on the ribosome

Our cryo-EM structure resolved NMT2 bound at the universal factor docking site near the polypeptide exit tunnel (Blau et al, 2005; Pech et al, 2010; Kramer et al, 2002) (Fig. 4A–E). Specifically, NMT2 was observed to contact uL23 via a highly conserved helix that acts as an anchor. This anchor helix (αE) contains several positively charged residues that establish electrostatic interactions

with uL23 (Fig. 4C,F). Notably, two salt bridges are formed between Arg324 (NMT2) and Glu91 (uL23) and between Arg330 (NMT2) and Asp145 (uL23), as suggested by the resolved side-chain density (Figs. 4C and EV3B). Several residues on this helix also contribute additional electrostatic contacts between NMT2 and the ribosome. Particularly, the NMT2 anchor helix interacts with a flipped-out rRNA nucleotide (G2711) at the base of H59 via Lys327 (Fig. 4D). A second contact between a separate NMT2 anchor helix (αD) and the ribosome is observed, with Arg306 stacking on a flipped nucleotide U2707 in H59 (Fig. 4E). These two flipped nucleotides in H59 appear to form an rRNA clamp that fixes NMT2 in place on the ribosome (Figs. 4B and EV4E).

Sequence alignments of NMT2 revealed that these contact points are conserved in higher eukaryotes, underscoring their importance in anchoring NMT2 to the ribosome (Fig. 4F). Furthermore, an alignment of human NMT2 with NMT1 showed conservation of the contact sites, suggesting that NMT isoforms

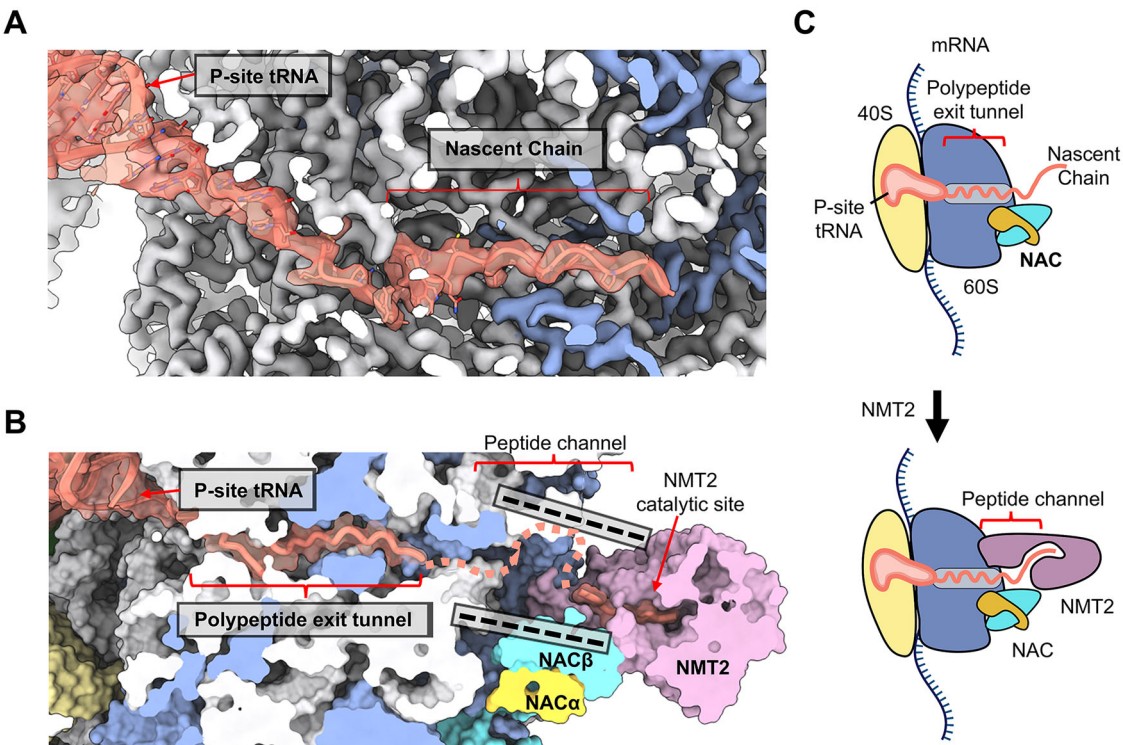

**Figure 3. NAC and NMT2 form an extended channel for the nascent chain from the polypeptide exit tunnel to the catalytic site of NMT2.**

(A) A close-up on the polypeptide exit tunnel (PET) showing a tRNA molecule stalled in the P-site of the ribosome, with a nascent chain EM-density extending from it. (B) Surface model of the extended channel formed by NAC (cyan, gold) and NMT2 (pink) from the polypeptide exit tunnel. (C) A schematic depicting how the nascent chain is exposed when only NAC is bound to the ribosome and is covered once NMT2 is recruited to form an extended channel.

share a similar mode of binding to the ribosome (Fig. 4F). To validate the importance of these contacts and assess their impact on NMT2 function, we performed site-directed mutagenesis to generate charge reversal mutants. These included a single mutant targeting electrostatic interactions with H59 (R306E), a double mutant disrupting both stacking with RNA H59 base G2711 as well as a conserved salt bridge (R324E/K327E), and a triple mutant combining disruptions of both H59 interactions as well as protein contacts (R306E/R324E/K327E). All three mutants exhibited a 2–3-fold reduction in NMT2 binding to the ribosome, both in the presence and absence of NAC (Fig. 4G–I; Appendix Fig. S2A,B). Collectively, these contacts enable NMT2 to dock near the polypeptide exit tunnel and engage nascent chain substrates efficiently for myristoylation.

## The NACβ C-tail and NMT2 N-tail mediate NMT2 binding to the ribosome

In the structure of the ternary complex, we were also able to observe NMT2 contacting NAC. In particular, a loop region of NMT2 interacts with the portion of NACβ that forms the barrel-like domain of the NAC heterodimer. This loop binds NAC via two histidine residues, His411 and His 412, which are in close proximity to His93 of NACβ (Fig. 5A). A sequence alignment demonstrates these residues are highly conserved across most eukaryotes (Fig. 5B). This region of NAC is not known to function in recruitment of other enzymes and is instead involved in the dimerization of NACα and NACβ subunits.

Therefore, this interaction between NAC and NMT2 may serve to orient the catalytic site of NMT2 above the polypeptide exit tunnel and would help stabilize NMT2 binding on the ribosome by providing an additional contact point. To investigate this further, we introduced two separate histidine-to-glutamic acid mutations into NMT2—H411E and H412E. The ability of these mutants to bind the ribosome was assessed using a co-sedimentation assay in the presence of NAC. The NMT2 H412E mutant showed a mild reduction in binding to the ribosome (Fig. 5C–E; Appendix Fig. S3A,B). Together, the co-sedimentation assays indicate that the NMT2 anchor on the ribosome is crucial for NMT2 binding, while the contacts with NAC function to stabilize NMT2 and the channel extension for nascent polypeptides to be guided to the catalytic site of NMT2.

Recent studies showed that NAC recruits various processing enzymes to the ribosome through distinct regions on the N- or C-terminal tails of either NACα or NACβ (Fig. 6A) (Gamerdinger et al, 2023; Lentzsch et al, 2024). The C-terminal arm of NACα contains a UBA domain (Spreter et al, 2005) involved in SRP binding (Jomaa et al, 2022), and the C-terminal arm of NACβ contains a patch of hydrophobic residues that recruits MAP1 via interaction with a zinc finger domain in the N-terminal extension region of MAP1 (Gamerdinger et al, 2023; Vetro and Chang, 2002). Therefore, we hypothesized that NAC can also recruit NMT2 to the ribosome using one of these flexible regions.

To biochemically test this, we truncated the flexible tails of NAC responsible for enzyme and targeting factor recruitment (NACα ΔN, NACα ΔC, and NACβ ΔC) to be used in co-sedimentation

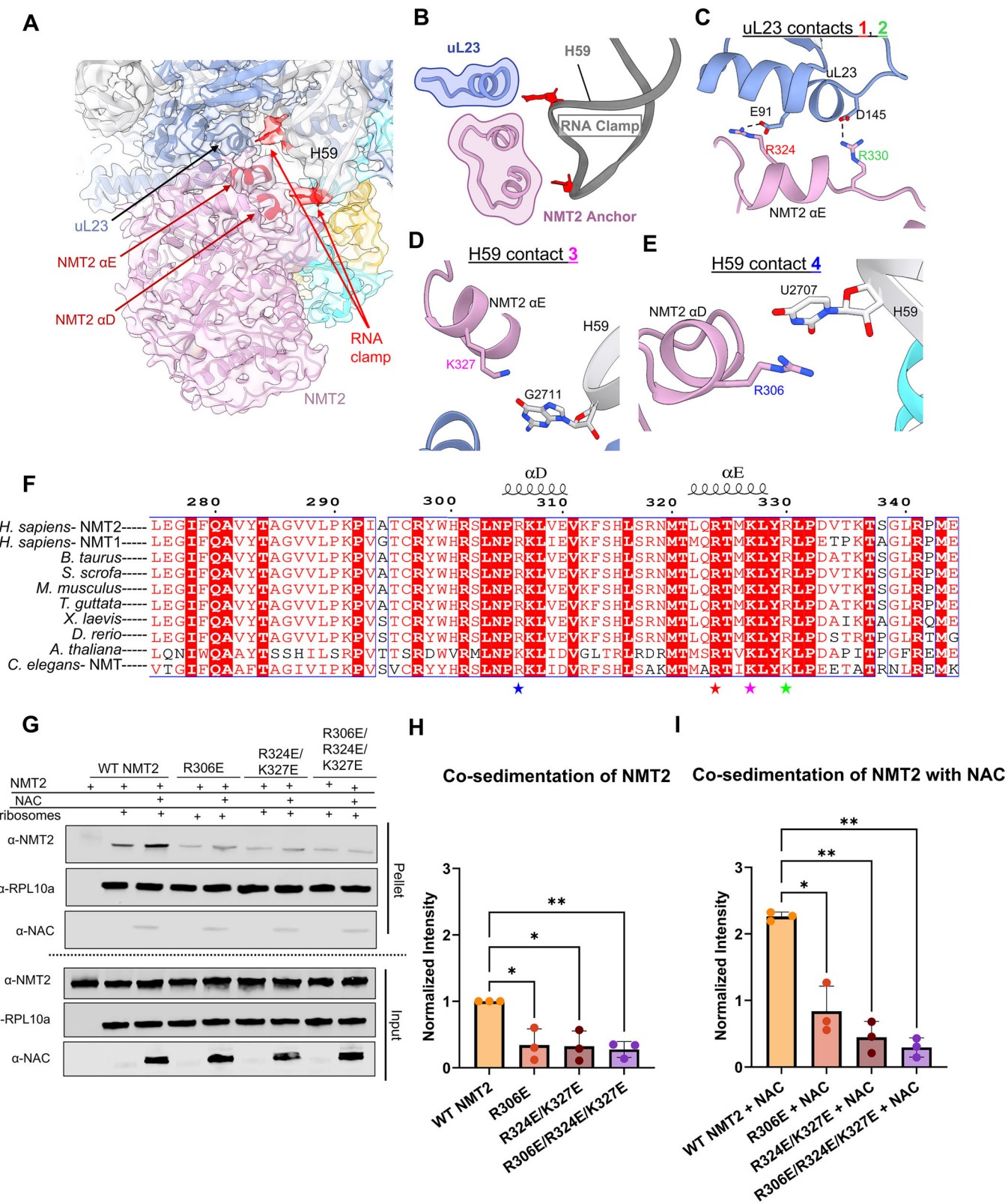

◄ **Figure 4. NMT2 contacts the ribosome via interactions with uL23 and H59.**

(A) Overview of the model depicting NMT2 contacts with the ribosome. H59 flipped out bases and the NMT2 anchor helices αD and αE are also indicated in red. Cryo-EM density is shown as a transparent surface and filtered to 4 Å. (B) Close-up of the model depicting the flipped-out bases G2711 and U2707 that form the rRNA clamp of NMT2. (C) Close-up of the NMT2 (pink) anchor helix, αE, forming two electrostatic contacts with uL23 (blue) using residues R324 (red) and R330 (green) (contacts 1 and 2) shown as cartoon. (D) A close-up of contact 3 depicting the residue on αE, K327 (magenta), making electrostatic contacts with H59 (gray) via a flipped base, G2711. (E) Close-up on the H59 contact made with residue R306 (blue) on the other anchor helix, αD, contacting U2707 (contact 4). (F) A sequence alignment between human NMT1 and NMT2, and bovine, pig, mouse, zebra finch, African clawed frog, zebrafish, and thale cress NMT2, and roundworm NMT highlights the conservation of positively charged residues at positions 306, 324, 327, and 330 (shown with blue, red, magenta, and green stars, respectively). Helices above the alignment indicate where αD and αE are in the sequence. The alignment was created using ESPript3.0 (ENDscript—https://endscript.ibcp.fr) (Robert and Gouet, 2014). (G) An immunoblot of the input and pellet fractions of a co-sedimentation assay carried out with WT NMT2, R306E, R324E/K327E, and R306E/R324E/K327E. (H) A quantification of co-sedimentation results from NMT2 alone lanes. Assays were completed as three independent replicates and band intensity was weighted by RPL10a band intensity per lane and normalized to WT NMT2. $P$ values are 0.0426 (R306E), 0.0372 (R324E/K327E), and 0.0091 (R306E/R324E/K327E), respectively. (I) A quantification of the co-sedimentation results from reactions containing both NMT2 and NAC shown in (F). Assays were completed as three independent replicates, and band intensity was weighted by RPL10a band intensity per lane and normalized to WT NMT2. $P$ values are 0.0295 (R306E + NAC), 0.0078 (R324E/K327E + NAC), and 0.0030 (R306E/R324E/K327E + NAC), respectively. Data information: In (H, I), data are presented as mean ± SD. $*p < 0.05$, $**p < 0.01$ (one-way ANOVA and uncorrected Fisher's LSD post hoc). Source data are available online for this figure.

assays (Fig. 6B; Appendix Fig. S4A). The truncations of the flexible tails did not impair NAC binding to ribosomes, which is mediated by the N-terminal anchor of NACβ (Fig. 6A,C). We then performed co-sedimentation assays between NMT2, ribosomes isolated from HEK293 cells, and the various NAC mutants (Fig. 6C; Appendix Fig. S4B,C). NMT2 fails to efficiently co-sediment with ribosomes in the presence of the NACβ ΔC tail mutant, instead pelleting at levels similar to NMT2 reacted with ribosomes only (Fig. 6D). In comparison, truncations of the NACα N- and C-tail did not affect NMT2 binding when compared to the control reaction containing both ribosomes and WT NAC.

The N-terminal tail of NMT2 contains an unstructured region that has previously been shown to be necessary for ribosomal localization (Glover et al, 1997). To investigate whether this region is also involved in NAC binding and subsequent recruitment to the ribosome, we generated an NMT2 ΔN mutant lacking the first 105 amino acids. We then performed a co-sedimentation assay of NMT2 ΔN in the presence or absence of WT NAC and NACβ ΔC (Fig. 6E; Appendix Fig. S4D,E). Our results show that the binding of NMT2 ΔN to the ribosome is decreased compared to WT NMT2 (Fig. 6F). Furthermore, reactions with WT NAC fail to rescue binding to levels observed for WT NMT2. Of note, NMT2 ΔN binding was further reduced when reacted with the NACβ ΔC-tail mutant compared to NMT2 ΔN in the absence of NAC (Fig. 6F). Together, these results show that the NACβ C-tail and the NMT2 N-tail mediate NMT2 binding to the ribosome.

## Discussion

Co-translational modification of nascent polypeptide chains is a complex process that is critical for protein biogenesis and requires the spatio-temporal coordination of many enzymes in order to generate properly modified proteins (Gamerdinger and Deuerling, 2024; Varland et al, 2015; Giglione et al, 2015). One such modification is the process of myristoylation by NMT1 or NMT2, which enables the protein to associate with membranes. Despite its fundamental importance in cell proteostasis, the mechanism by which NMT2 interacts with the ribosome and whether NAC plays a role in this process remains poorly understood.

While previous studies show that NAC binds all translating ribosomes and is the first factor nascent proteins encounter when exiting the large subunit (del Alamo et al, 2011; Raue et al, 2007), our results now reveal its regulatory role in NMT2 recruitment and ribosomal binding and allow us to propose the following model. When a myristoylation target emerges, the C-terminal tail of NACβ recruits or enhances NMT2 binding to the ribosome in a similar manner as MAP1 (Gamerdinger et al, 2023). NAC orients NMT2 on the ribosome, with the NAC-NMT2 loop-barrel contact forming to stabilize the ternary complex. The interaction between NAC and NMT2 extends the polypeptide exit tunnel, generating an extended peptide channel to the catalytic core of NMT2 and positioning the catalytic site to receive the emerging nascent chain. NMT2 will then engage the exposed GXXXKS motif for co-translational N-myristoylation to occur (Fig. 7).

NMT2 makes direct contact with the ribosome utilizing residues of two highly conserved anchor helices that connect the two GNAT domains of NMT2 to one another (Dian et al, 2020). NMT2 binds to both H59 and uL23 on the ribosome, in the universal factor docking site (Kramer et al, 2002). This docking position uses an RNA clamp from H59 as well as electrostatic interactions with uL23 to keep NMT2 properly positioned near the polypeptide exit tunnel. When comparing the binding site of NMT2 to MAP1 (Fig. EV5A,B), we notice significant binding site overlaps that would prevent these proteins from being bound to the ribosome together. This supports our model of sequential co-translational processing. MAP1 is responsible for initiator methionine cleavage (Bradshaw et al, 1998), which N-myristoylation targets need to undergo to expose the proper myristoylation motif on nascent chains prior to NMT2 recruitment and binding. For this to happen, MAP1 must bind the ribosome and then unbind prior to NMT2 recruitment to allow for subsequent NMT2 binding. The key ribosomal contact residues of NMT2 are conserved in NMT1 as well, indicating that the mode of NMT1 docking on the ribosome may be shared between the homologs.

Our co-sedimentation assays also demonstrate that the C-terminal tail of NACβ, which has been previously shown to recruit MAP1 to the ribosome (Gamerdinger et al, 2023), is also responsible for NMT2 recruitment to the ribosome. This provides additional support for our model of sequential co-translational modification in the case of N-glycine myristoylation. Furthermore, our results suggest that the NACβ C-terminal tail interacts with the N-terminal tail of NMT2. The NMT2 tail can also independently enhance interactions with the ribosome, as its deletion exhibited reduced binding in our biochemical assays.

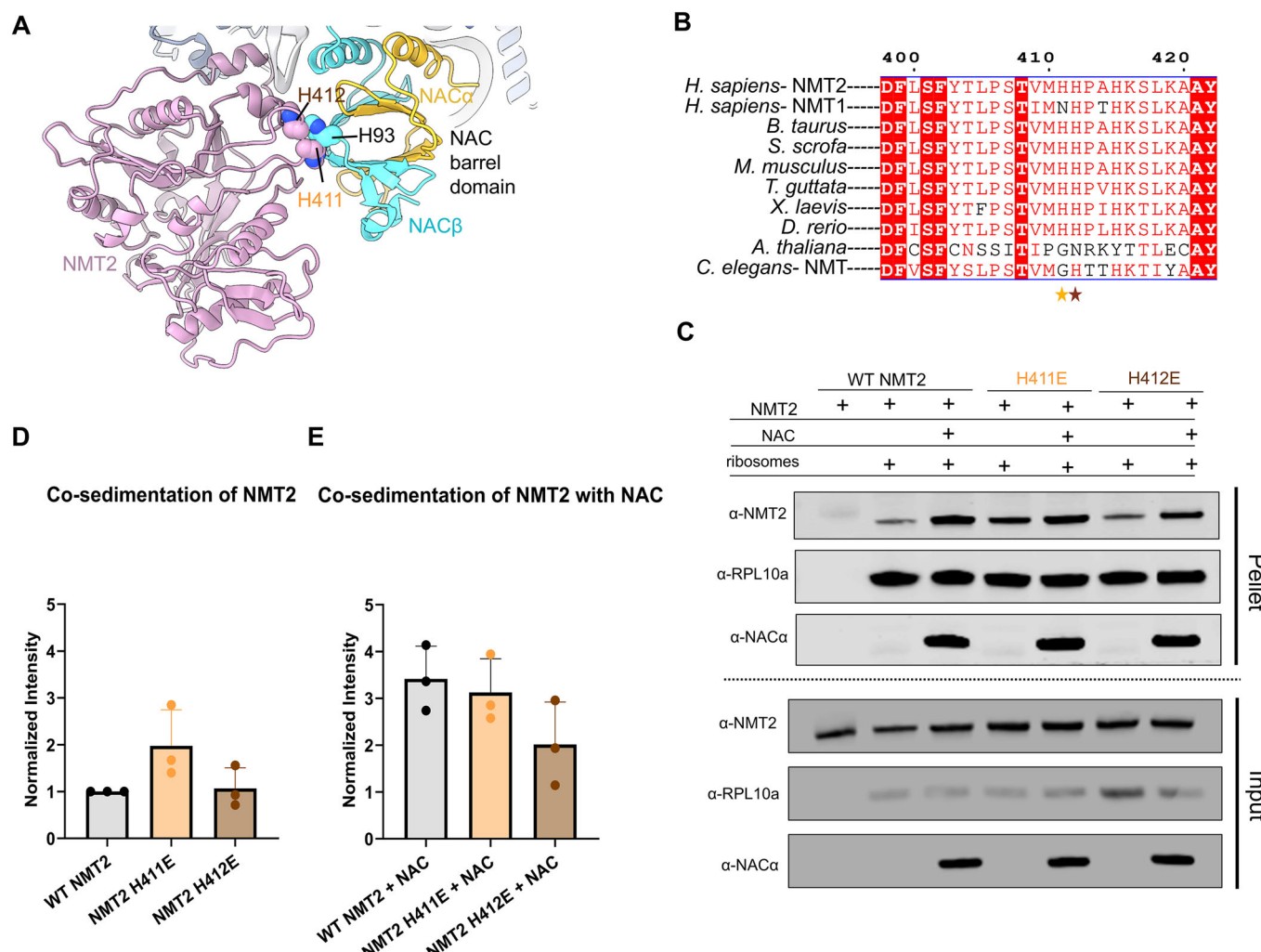

**Figure 5. NAC orients NMT2 on the ribosome via its barrel domain.**

(A) The structure of NMT2 (pink) in complex with the ribosome and NAC (gold, cyan) depicting the location of the histidine residues contacting one another in the loop-barrel contact formed between NMT2 and NAC. (B) A sequence alignment between human NMT1 and NMT2, and bovine, pig, mouse, zebra finch, African clawed frog, zebrafish, and thale cress NMT2, and roundworm NMT. H411 is indicated with an orange star, H412 is indicated with a brown star. (C) An immunoblot of both input and pellet fractions of a co-sedimentation assay showing NMT2 recruitment to the ribosome for WT NMT2, H411E (orange), and H412E (brown) both in the presence and absence of NAC. (D) A quantification of co-sedimentation results from NMT2 alone lanes of the blot shown in (C). Assays were completed as 3 independent replicates, and band intensity was weighted by RPL10a band intensity per lane and normalized to WT NMT2. Both H411E and H412E are not significant after analysis, with p-values of 0.1593 (H411E) and 0.8152 (H412E), respectively. (E) A quantification of the co-sedimentation results from reactions containing both NMT2 and NAC shown in (C). Assays were completed as three independent replicates and band intensity was weighted by RPL10a band intensity per lane and normalized to WT NMT2. Both H411E and H412E are not significant after analysis, with p values of 0.3857 (H411E) and 0.0803 (H412E), respectively. Data information: In (D, E), data are presented as mean ± SD. One-way ANOVA and uncorrected Fisher's LSD post hoc test were used to determine p values. Source data are available online for this figure.

NAC and NMT2 also interact via a loop region in NMT2 and the barrel-like dimerization domain of NACβ. This domain of NAC is anchored to the ribosome, and because of its low sampling space, likely does not aid in the recruitment of enzymes to the polypeptide exit tunnel. Instead, we hypothesize that this interaction serves to stabilize NMT2 on the ribosome in a conformation that allows for the most efficient engagement of the nascent chain as it exits the ribosome. This hypothesis was supported by both our structural observations and our biochemical data showing that binding of NMT2 to the ribosome is not significantly decreased in the presence of either histidine mutant. The sequence alignment between NMT2 and NMT1 shows a loss of the histidine residue

at position 411 within the loop region of NMT1 (Fig. 5B), so further exploration would be needed to determine if the stabilizing interaction is maintained between the two isoforms.

The structural analysis of the RNC$_{MARCKS}$:NMT2:NAC ternary complex, combined with our biochemical experiments, determined the molecular mechanism of the initial steps of nascent chain processing by NMT2. This work advances our understanding of nascent chain co-translational processing dynamics and the interplay of different factors competing for overlapping binding sites on translating ribosomes. Dysregulation of NMT2, but not NMT1, has been linked to heart failure and cardiac hypertrophy (Tomita et al, 2023). Future investigation is needed to understand how NMT2

exclusively selects its substrates on the ribosome, which will help develop better strategies for understanding the etiology of these diseases.

# Methods

### Reagents and tools table

| Reagent/resource | Reference or source | Identifier or catalog number |
|---|---|---|
| **Experimental models** | | |
| HEK 293T | ATCC | CRL-3216 |
| HEK 293 GnTI | ATCC | CRL-3022 |
| *E. coli* BL21-CodonPlus DE3 | Agilent | 230245 |
| **Recombinant DNA** | | |
| pET-28a(+)-NMT2 | GenScript Plasmid Synthesis | |
| pET28-6xHis-NACa-NACb | Jomaa et al (2022) | |
| **Antibodies** | | |
| Mouse anti-FLAG | Sigma | F1804-50UG |
| Rabbit anti-NACα | Biorbyt | orb411671 |
| Rabbit anti-NACβ | Invitrogen | PA5-63299 |
| Rabbit anti-NMT2 | ThermoFisher | PA5-144212 |
| Rabbit anti-RPL10a | Invitrogen | MA5-44710 |
| Goat anti-Mouse | Invitrogen | A21058 |
| Goat anti-Rabbit | Invitrogen | A32735 |
| **Oligonucleotides and other sequence-based reagents** | | |
| MARCKS nascent chain construct | This study | See "Methods" for sequence |
| **Chemicals, enzymes, and other reagents** | | |
| Protease inhibitor cocktail | Promega | G6521 |
| RNase inhibitor | ThermoFisher | EO0381 |
| QIAquick PCR Purification Kit | Qiagen | 28104 |
| HiScribe T7 High Yield RNA Synthesis Kit | NEB | E2040S |
| Flexi Rabbit Reticulocyte Lysate System | Promega | L4540 |
| SurePAGE 4–12% Bis-Tris gels | Genscript | M00653, M00654 |
| MES 1× running buffer | Genscript | M00677 |
| Myristoyl Coenzyme A | Cayman Chemicals | 36777 |
| **Software** | | |
| CryoSPARC v4.6.0 | https://cryosparc.com/ Punjani et al (2017) | |
| PHENIX v.1.20.1 | https://phenix-online.org/ Liebschner et al (2019) | |
| Coot v.0.9.8.93 | https://www2.mrc-lmb.cam.ac.uk/personal/pemsley/coot/ Emsley P, Lohkamp B, Scott WG, Cowtan K (2010) | |
| ChimeraX v1.6.1 | https://www.cgl.ucsf.edu/chimerax/ Goddard TD, Huang CC, Meng EC, Pettersen EF, Couch GS, Morris JH, Ferrin TE (2018) | |
| ESPript 3.0 | https://espript.ibcp.fr/ESPript/cgi-bin/ESPript.cgi Robert and Gouet (2014) | |
| ClustalOmega | https://www.ebi.ac.uk/jdispatcher/msa/clustalo Madeira F, Madhusoodanan N, Lee J et al (2024) | |
| GraphPad Prism | https://www.graphpad.com/ | |
| ImageJ | https://imagej.net/ij/ | |
| **Other** | | |
| 1 mL polypropylene gravity column | Qiagen | 34924 |
| Anti-DYKDDDK affinity resin | GenScript | L00432 |
| DYKDDDK peptide | GenScript | RP10586-1 |
| Nitrocellulose membrane | LI-COR | 926-31092 |
| Quantifoil R2/1 300 mesh copper grids | Quantifoil | Q350CR1 |
| Glow discharger | EMS | |
| Titan Krios | ThermoFisher | |

### Polysome analysis by sucrose gradient centrifugation

Polysome profiling of cells over sucrose gradients was performed using the procedure as described in a previous study (Karamysheva et al, 2018). Briefly, a 100 mg pellet of HEK293T cells was lysed in 500 μL of lysis buffer (20 mM HEPES-KOH pH 7.4, 100 mM KCl, 5 mM MgCl₂, 0.5% IGEPAL, 1 mM DTT, 0.5× protease inhibitor cocktail (Promega REF G6521), 4 units/mL RNase inhibitor (ThermoFisher REF EO0381), and 100 μg/mL cycloheximide). The lysate was passed through a 23 G needle six times and then centrifuged for 8 min at 4 °C with $11{,}200 \times g$ to clear the debris. The resulting clarified lysate was diluted to an $A_{260}$ value of 30 units (500 μL) and was loaded onto a continuous 10–50% (w/v) sucrose gradient prepared in a buffer containing 20 mM HEPES-KOH (pH 7.4), 100 mM KCl, 10 mM MgCl₂, and 1 mM DTT. The gradient was centrifuged at $230{,}000 \times g$ for 2.5 h at 4 °C using an SW40 rotor. Polysome profile was then generated by fractionating the gradient at 0.2 mL/min into 800 μL fractions using a BioComp gradient fractionator. Absorbance at 260 nm was measured continuously during fractionation to monitor RNA distribution. A 500 μL volume of each fraction was precipitated

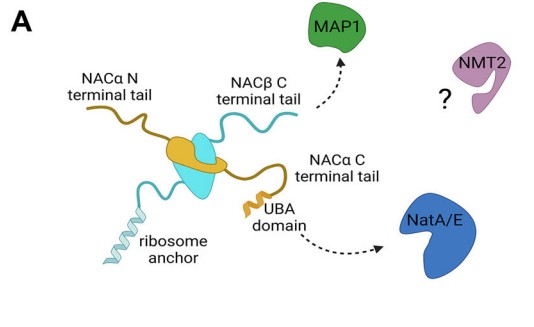

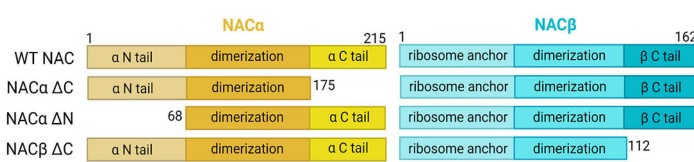

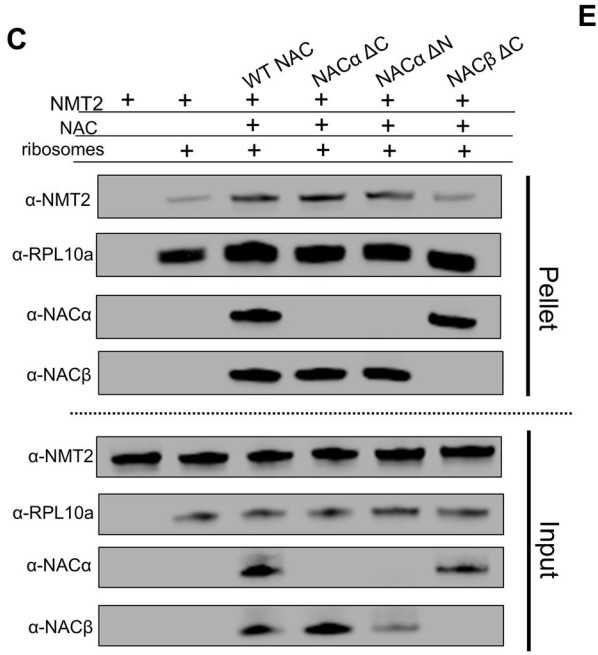

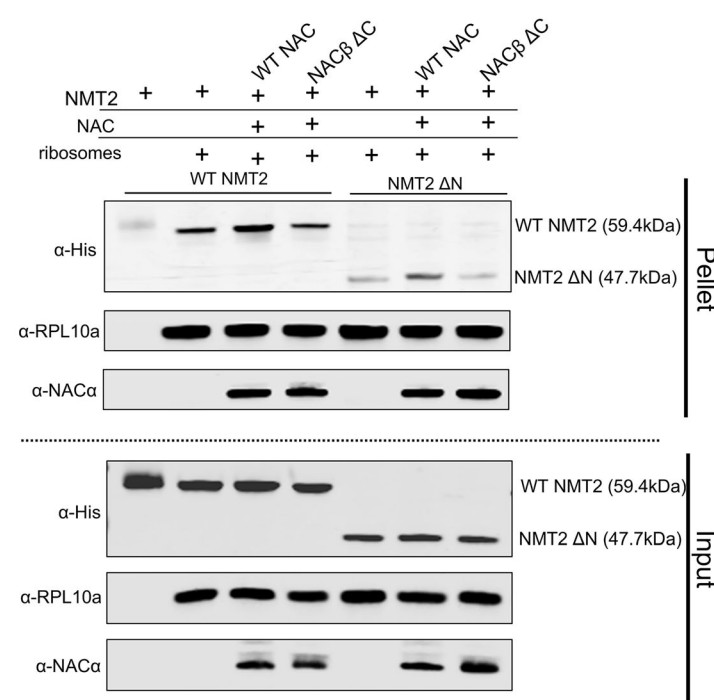

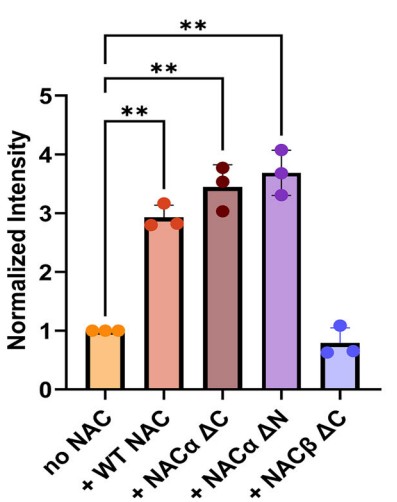

**D**

**Co-sedimentation of NMT2 with NAC mutants**

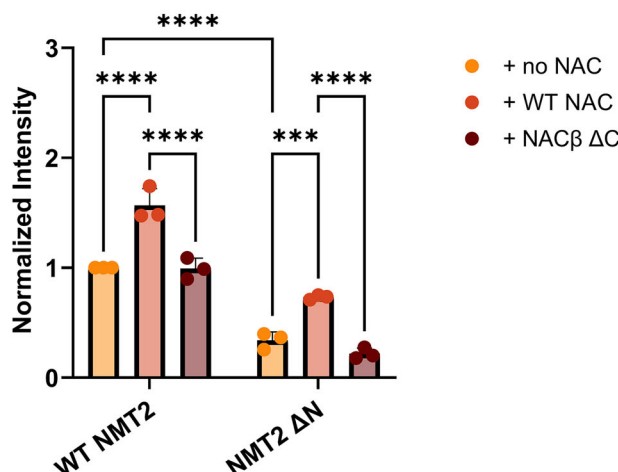

**F**

**Co-sedimentation of NMT2 and NMT2 ΔN**

**Figure 6. Tail regions of NAC and NMT2 are important for NMT2 binding and recruitment to the ribosome.**

(A) A schematic of NAC depicting the dimerization domains and N- and C-terminal tails of both the α subunit (gold) and the β subunit (cyan). The C terminal tail of NACβ has been shown to recruit MAP1 (green), and the C terminal tail of NACα has been shown to recruit NatA/E (blue). (B) A schematic of the NAC tail mutants generated. NACα ΔC, NACα ΔN, and NACβ ΔC have truncated tail regions but preserve both dimerization domains and the ribosome anchor at the N terminus of NACβ. (C) An immunoblot of both input and pellet fractions of a co-sedimentation assay performed between WT NMT2 and various NAC full length and truncated constructs. Both NACα and NACβ antibodies are used to blot for NAC in the case of the truncated mutants. (D) A quantification of the co-sedimentation results from the co-sedimentation shown in (C). Assays were completed as three independent replicates, and band intensity was weighted by RPL10a band intensity per lane and normalized to WT NMT2, and significance was determined using a one-way ANOVA, and uncorrected Fisher's LSD post hoc test. *P* values are 0.0038 (WT NAC), 0.0079 (NACα ΔC), 0.0068 (NACα ΔN), and 0.2936 (NACβ ΔC), respectively. (E) An immunoblot of both input and pellet fractions of a co-sedimentation assay performed using WT NMT2 and NMT2 ΔN with NAC and NACβ ΔC. NMT2 constructs contain an N-terminal 6X His-tag that was detected via immunoblotting. (F) A quantification of the co-sedimentation results from the co-sedimentation shown in (D). Assays were completed as three independent replicates, and band intensity was weighted by RPL10a band intensity per lane and normalized to WT NMT2, and significance was determined using a two-way ANOVA, and Tukey's multiple comparisons post hoc test. All pairwise comparisons were significant besides WT NMT2 alone vs. WT NMT2 + NACβ ΔC ($p = 0.9930$), and NMT2ΔN alone vs. NMT2ΔN + NACβ ΔC ($p = 0.2122$). *P* values for all other comparisons are <0.0001, except for NMT2ΔN alone vs. NMT2ΔN + WT NAC ($p = 0.0002$). Data information: In (D, F), data are presented as mean ± SD. *$p < 0.05$, **$p < 0.01$, ***$p < 0.001$, ****$p < 0.0001$. Source data are available online for this figure.

with TCA at a final concentration of 10% (w/v) by incubating on ice for 30 min. Precipitated protein fractions were resuspended in 2X SDS-PAGE loading buffer (100 mM Tris-HCl pH 6.8, 4% SDS, 12% glycerol, 0.008% bromophenol blue, 0.2 M DTT) before running a gel and immunoblotting (see below methods).

## Protein purification—NMT2 and mutants

N-terminal 6x His-TEV tagged human NMT2 cloned into pET-28a(+) vector was expressed in Escherichia coli BL21 pRIL cells. The cells were grown to an $OD_{600}$ of ~0.6–0.8 before induction of NMT2 expression with 1 mM IPTG, followed by overnight growth at 18 °C. A total of 5 g of cell pellet was harvested from a 2 L culture. The cell pellet was resuspended in lysis buffer (50 mM HEPES pH 7.5, 500 mM NaCl, 10% glycerol, 5 mM BME, 0.5× protease inhibitor cocktail) and lysed using one passage through a French press at 10,000 PSI. The supernatant of the final spin containing soluble NMT2 was loaded onto a nickel column for subsequent purification via immobilized metal affinity chromatography. The flowthrough of the column load was collected, then the column was washed with 50 mL of nickel wash buffer (50 mM HEPES pH 7.5, 500 mM NaCl, 50 mM Imidazole, 10% glycerol, 5 mM BME). Protein was eluted off the column with a stepwise elution of 15%, 30%, and 80% concentration of nickel elution buffer (50 mM HEPES pH 7.5, 500 mM NaCl, 300 mM Imidazole, 10% glycerol, 5 mM BME), and 1.2 mL fractions were collected. Fractions containing NMT2 were pooled and dialyzed overnight at 4 °C with TEV in order to cleave the N-terminal 6x His tag in dialysis buffer (50 mM HEPES pH 8.0, 50 mM NaCl, 10% glycerol, 5 mM BME). After dialysis, NMT2 was further purified using a HiTrap Q anion exchange column. The column was equilibrated with 5% concentration of high salt buffer (50 mM HEPES, pH 8.0, 1 M NaCl, 10% glycerol, 2 mM DTT) and the protein was loaded onto the column with the flowthrough collected. The FPLC was equilibrated with 95% of no salt buffer (50 mM HEPES, pH 8.0, 10% glycerol, 2 mM DTT) and 5% of high salt buffer (50 mM HEPES, pH 8.0, 1 M NaCl, 10% glycerol, 2 mM DTT). The protein was subjected to a linear gradient from 5% to 100% high salt buffer and eluted in a single peak with approximately 300 mM NaCl. The collected fractions containing NMT2 were pooled, spin-concentrated, and buffer-exchanged into NMT2 storage buffer (50 mM HEPES pH 7.5, 150 mM KCl, 10% glycerol, 2 mM DTT) for use in later experimentation.

## Protein purification—NAC and mutants

6xHis-NACα and NACβ were co-expressed in *Escherichia coli* BL21-CodonPlus DE3 competent cells. The cells were grown to an $OD_{600}$ of ~0.6-0.8 at 37 °C and induced with 1 mM IPTG overnight at 18 °C. A cell pellet was harvested and resuspended in lysis buffer (50 mM HEPES pH 7.5, 1 M NaCl, 10% glycerol, 6 mM BME, protease inhibitor cocktail (Roche)) and lysed using two passages through a French press at 10,000 PSI. The lysate was spun down at $36,355 \times g$ for 30 min, 4 °C using a TI 50.2 rotor (2X). The supernatant was loaded onto a nickel column. The column was washed with 50 mL of nickel wash buffer (50 mM HEPES pH 7.5, 1 M NaCl, 45 mM Imidazole, 10% glycerol, 6 mM BME). Protein was eluted off the column with a stepwise elution of 15%, 30%, and 80% concentration of nickel elution buffer (50 mM HEPES pH 7.5, 150 mM KOac, 300 mM Imidazole, 10% glycerol, 6 mM BME), and 1.2 mL fractions were collected. Fractions containing NAC were pooled and dialyzed overnight at 4 °C with 3C protease in order to cleave the N-terminal 6x His tag in dialysis buffer (50 mM HEPES pH 7.5, 150 mM KOac, 10% glycerol, 6 mM BME). Dialyzed NAC was then loaded onto a HiTrap Q anion exchange column. The column and FPLC were equilibrated with no salt buffer (50 mM HEPES, pH 7.5, 10% glycerol, 2 mM DTT, 2 mM EDTA). The protein was loaded onto the column and was subjected to a linear gradient from 5% to 100% high salt buffer (50 mM HEPES, pH 7.5, 1 M NaCl, 10% glycerol, 2 mM DTT, 2 mM EDTA) and eluted with around 30% high salt buffer. The collected fractions containing NAC were pooled, spin-concentrated, and buffer-exchanged into storage buffer (50 mM HEPES pH 7.5, 150 NaCl, 10% glycerol, 2 mM DTT).

## Isolation of salt-washed ribosomes from HEK293 cells

HEK293 GnTI cell pellets were used for ribosome isolation. The cell pellet was resuspended in lysis buffer (50 mM HEPES KOH pH 7.7, 750 mM KOAc, 5 mM Mg(OAc)$_2$, 0.5% IGEPAL, 1 mM DTT, 0.5× Protease Inhibitor (Promega REF G6521), 0.04 U/μL RNase Inhibitor (ThermoFisher REF EO0381), 100 μg/mL Cycloheximide), and 2 mL of buffer was used per 1 g of pellet. After an initial resuspension by gentle pipetting, the cells were lysed by passing the solution through a syringe with a 23 G × 1½ (0.6 mm × 40 mm) needle (BD PrecisionGlide Needle REF 305194) a minimum of 8 times and were incubated on ice for a minimum of 1 h. The lysate

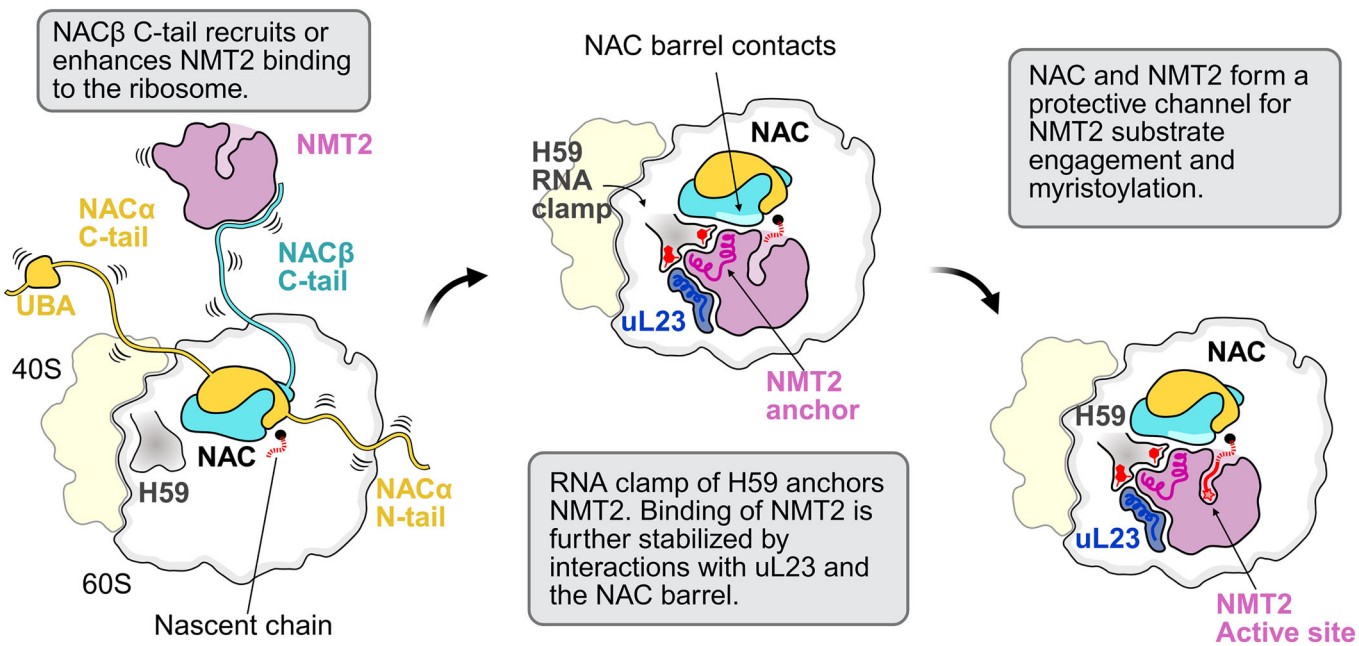

**Figure 7. Model for co-translational myristoylation by NMT2 on the ribosome.**

NMT2 binding to the ribosome is mediated by contacts with the NACβ C-tail. NMT2 is also anchored to the ribosome surface by the H59 rRNA clamp and is oriented for efficient substrate engagement by the contacts made with the NAC barrel. The conformation of NAC and NMT2 on the ribosome allows them to form a channel from the polypeptide exit tunnel that guides emerging nascent chains into the NMT2 active site for substrate myristoylation.

was transferred to a sterile 1.5 mL tube and was centrifuged at 11,000 × $g$ for 10 min at 4 °C. The supernatant of that spin was taken and layered over an equivalent volume of 30% sucrose buffer (30% sucrose, 50 mM HEPES KOH pH 7.7, 750 mM KOAc, 5 mM Mg(OAc)$_2$, 1 mM DTT) in 10.4 mL polycarbonate bottles (Beckman Coulter Product No. 355651) and spun overnight at 116,000 × $g$ in a Ti90 rotor (Beckman Coulter Product No. 355530). After the spin, the supernatant was removed and discarded, and the pellet was resuspended in resuspension buffer at an A$_{260}$ value of 400 units (50 mM HEPES KOH pH 7.7, 100 mM KOAc, 15 mM Mg(OAc)$_2$, 0.5× Protease Inhibitor, 0.04 U/μL RNase Inhibitor). The ribosomes were aliquoted, snap frozen, and stored at −80 °C for future experiments.

## PCR and in vitro transcription

Varying length constructs of MARCKS all containing an Xbp1 stalling sequence (Calfon et al, 2002; Shanmuganathan et al, 2019; Yoshida et al, 2001) were PCR amplified, including the T7 promoter sequence, N-terminal 3× FLAG tag, and 3C cleavage site. The final PCR products were obtained using a QIAquick PCR Purification Kit (Qiagen, 28104) and were in vitro transcribed using either a HiScribe T7 High Yield RNA Synthesis Kit (NEB, E2040S) or with 1.1 mg/mL T7 polymerase in reaction buffer (40 mM Tris-HCl pH 7.6, 5 mM ribonucleoside triphosphates, 6 mM MgCl2, 2 mM spermidine, 1 mM DTT, 0.04 U/μL RNase Inhibitor (ThermoFisher REF EO0381)). The reactions were incubated for 4 h at 38 °C and an equal volume of 6 M LiCl was added and incubated for 1 h on ice. The reactions were centrifuged at 24,147 × $g$ for 20 min at 4 °C in a Microliter 30 × 2 mL fixed-angle rotor

(ThermoFisher REF 75003652) and the supernatant was discarded. The pellet was washed with 200 μL of 70% EtOH and centrifuged again for another 5 min at 24,147 × $g$. The supernatant was once again discarded, and the pellet was resuspended in 100 μL of cold, sterile H2O. 40 μL of 2.8 M NaOAc and 300 μL of 95% EtOH were added to the resuspended pellet and pipetted gently to mix and left on ice for 5 min. The reaction was centrifuged at 24,147 × $g$ for an additional 30 min at 4 °C. The pellet was washed with 200 μL of 70% EtOH, centrifuged at 24,147 × $g$ at 4 °C for 5 min, air dried briefly, and resuspended in 100 μL of cold, sterile H2O.

## NMT2 co-sedimentation with varying lengths of RNC$_{MARCKS}$ nascent chain constructs

The crude RNCs were generated by reacting RRL from a Flexi Rabbit Reticulocyte Lysate System (Promega, L4540) with mRNA synthesized by in vitro transcription as described above. For the in vitro translation reaction to generate the stalled RNCs, 100 μg of RNA was added to 200 μL of RRL and supplemented with 0.04 U/μL RNase Inhibitor (ThermoFisher REF EO0381), 0.5× Protease Inhibitor (Promega REF G6521), 24 μM amino acid mix, 81 μM KCl, and 2 mM Mg(OAc)$_2$. The reaction was incubated for 25 min at 32 °C, then placed on ice. The reactions were then spun at 20,821 × $g$ at 4 °C for 10 min in a Microliter 30 × 2 mL fixed-angle rotor. The supernatant was loaded over an equal volume 30% sucrose cushion (50 mM HEPES-KOH pH 7.7, 100 mM KOAc, 15 mM Mg(OAc)$_2$, 30% sucrose) and spun in a TLA 120.1 rotor for 1 h at 4 °C and 355,040 × $g$. The pellet was resuspended in 100 μL of RNC buffer (50 mM HEPES-KOH pH 7.7, 100 mM KOAc, 15 mM Mg(OAc)$_2$), loaded onto another 30% sucrose cushion of 200 μL,

and spun again at $355,040 \times g$ at 4 °C for 1 h in a TLA 120.1 rotor. The supernatant was discarded, and the pellet was resuspended in 20 µL of RNC buffer.

NAC and NMT2 were diluted to 20 µM and were incubated on ice for 30 min in ribosome buffer (50 mM HEPES pH 7.7, 200 mM KOAc, 15 mM Mg(OAc)$_2$, 0.02% C12E8, 0.04 U/µL RNAse inhibitor (ThermoFisher REF EO0381)). The RNCs were diluted to an $A_{260}$ value of 4 units using the aliquot of ribosome buffer. Then, reactions containing 40 µL of RNCs were treated with 0.5 µL of 0.6 IU/µL of 3C protease to expose the N-terminal glycine of the nascent chains and were incubated for 20 min at 30 °C to allow cleavage to occur. NAC and NMT2 were added at a final concentration of 1 µM and the reactions were then allowed to incubate an additional 20 min at 30 °C. The reactions were cooled for 10 min on ice and 25 µL of the total mixture was taken and layered over 50 µL of sucrose buffer (50 mM HEPES pH 7.7, 200 mM KOAc, 15 mM Mg(OAc)$_2$, 30% sucrose, 0.04 U/µL RNAse inhibitor) in 0.5 mL ultracentrifuge tubes. 30 µL of input reaction were mixed with 3.75 µL of 5X SDS loading dye. The reactions were centrifuged at $355,040 \times g$ for 1 hr at 4 °C using a TLA 120.1 rotor. Following the spin, the supernatant was discarded, and the pellets were resuspended in 25 µL of 1× SDS loading dye. The input and pellet samples were run on a 4–12% gradient Bis-Tris gel, and the results were visualized with an immunoblot.

## NMT2 co-sedimentation assay with salt-washed ribosomes

NAC and NMT2 stocks were diluted in ribosome buffer (50 mM HEPES pH 7.7, 200 mM KOAc, 15 mM Mg(OAc)$_2$, 0.02% C12E8) to 10 µM and allowed to incubate on ice for approximately 30 min. The salt-washed ribosomes were diluted to an $A_{260}$ value of 13 units using ribosome resuspension buffer. Then, reactions containing 80 µL of diluted ribosomes and 200 nM final concentration of NAC and NMT2 were mixed and allowed to incubate at 30 °C for 20 min. The reactions were cooled for 10 min on ice and 50 µL of the total mixture was taken and layered over 100 µL of sucrose buffer (50 mM HEPES pH 7.7, 200 mM KOAc, 15 mM Mg(OAc)$_2$, 30% sucrose, 0.04 U/µL RNAse inhibitor (ThermoFisher REF EO0381)) in 0.5 mL ultracentrifuge tubes. The remaining reactions were centrifuged at $355,040 \times g$ for 1 h at 4C using a TLA 120.1 rotor. Immediately following the spin, the supernatant was decanted, and the ribosome pellets were resuspended in 25 µL of 1× SDS loading dye, pipetting up and down gently to avoid bubbles. The input and pellet samples were run on a 4–12% gradient Bis-Tris gel, and the results were visualized via immunoblot.

## Immunoblotting

Samples were run on SurePAGE 4–12% Bis-Tris gels (Genscript, M00653, M00654) in an MES 1× running buffer (Genscript, M00677) at 200 V for approximately 30 min, then transferred to a 0.2 µm nitrocellulose membrane (LI-COR, 926-31092). Prior to the addition of any antibodies, the membrane was blocked using 5% milk in 1× PBST for 1 h at room temperature with agitation. The membrane was washed using 1× PBST and antibodies in 3% BSA in 1× PBST were added for 1 h at room temperature, or overnight at 4 °C with agitation. Primary antibodies used were as follows: FLAG (Sigma, F1804-50UG), NACα (Biorbyt, orb411671), NACβ (Invitrogen, PA5-

63299), RPL10a (Invitrogen, MA5-44710), and NMT2 (Thermo-Fisher, PA5-144212). After incubation with primary antibody, the membrane was washed again with 1× PBST and was incubated again with a secondary antibody: anti-mouse (Invitrogen, A21058) or anti-rabbit (Invitrogen, A32735). The membrane was imaged on a LI-COR Odyssey imager to detect bands. Quantification of NMT2/NAC co-sedimentation assays was performed in triplicate. Band intensities from immunoblots were measured in ImageJ, with NMT2 intensities weighted by RPL10a loading per lane. Weighted NMT2 intensity was then normalized against WT NMT2 alone band intensity. The intensity data was first assessed for normality using the Shapiro–Wilk test, then statistics on band intensity changes were done using either a paired $t$ test for 2 samples, or a one-way ANOVA and Fisher's LSD post hoc test for experiments with >2 experimental groups. For the immunoblot quantification of NMT2 ΔN and NAC tail mutants where two variables are being assessed, a two-way ANOVA was carried out. The post hoc pairwise significance was determined using Tukey's multiple comparisons test. Significance between the groups is denoted for p-values of <0.05 (*), <0.01 (**), <0.001 (***), and <0.0001 (****).

## In vitro translation and RNC$_{MARCKS}$ purification for cryo-EM

The mRNA generated during the in vitro transcription step was reacted with RRLs using a Flexi Rabbit Reticulocyte Lysate System (Promega, REF L4540) for generation of a nascent chain with the following sequence: MDYKDHDGDYKDHDIDYKDDDDKSSG<u>LE VLFQGA</u>**QFSKTAAKGEAAAERPGEAAVASSPSKANGQENGH VKVNGDASPAAAE***DPVPYQPPFLCQWGRHQCAWKPLM* (3C cut site is underlined, MARCKS sequence is bolded) containing an N-terminal 3X FLAG tag and C-terminal Xbp1 stalling sequence (italicized). Briefly, 200 µg of RNA was added to 450 µL of RRL, and supplemented with 0.04 U/µL RNase Inhibitor (ThermoFisher REF EO0381), 0.5× Protease Inhibitor (Promega REF G6521), 24 µM amino acid mix, 81 µM KCl, and 2 mM Mg(OAc)$_2$. Once combined, the reaction was incubated for 25 min at 32 °C, then placed on ice.

A 1 mL polypropylene gravity column (Qiagen, 34924) was rinsed with 4 mL of 1× PBS and then 200 µL of Anti-DYKDDDK G1 affinity resin (GenScript, L00432) slurry was loaded onto the column. The resin was washed with 10 column volumes of 1× PBS and 10 column volumes of wash buffer 1 (50 mM HEPES-KOH pH 7.7, 100 mM KCl, 100 mM MgCl$_2$). After the 25-min incubation at 32 °C and transfer to ice, the previously prepared in vitro translation reaction was loaded onto the column, which was sealed with parafilm and incubated for 2 h at 4 °C with constant rotation. After 2 h, the column was washed with 10 column volumes of wash buffer 2 (50 mM HEPES-KOH pH 7.7, 500 mM KCl, 10 mM MgCl$_2$, 0.1% Triton-X), followed by 10 column volumes of wash buffer 1. To elute the RNCs from the column, a 1.2 mL solution of 0.25 mg/mL DYKDDDK peptide (GenScript, RP10586-1) was prepared using elution buffer (50 mM HEPES-KOH pH 7.7, 100 mM KOAc, 15 mM Mg(OAc)$_2$) to dilute the peptide, the column was moved to room temperature, and 110 µL of DYKDDDK peptide was added to the beads. This was allowed to incubate at room temperature for 15 min before eluting and was repeated 4 times. The elution fractions were then pooled and centrifuged at $88,760 \times g$ at 4 °C for 90 min using a TLA 120.1 rotor. The supernatant was discarded, and the pellet was resuspended in

elution buffer by gently pipetting. The ribosomes were then used immediately for cryo-EM sample preparation.

## RNC_MARCKS:NMT2:NAC cryo-EM sample preparation and imaging

Previously prepared RNC_MARCKS were diluted to 250 ng/μL using cryo-EM buffer (50 mM HEPES KOH pH 7.7, 80 mM KOAc, 10 mM Mg(OAc)$_2$, 0.05% C12E8) and were first treated with 3C protease, RNase inhibitor was added and was incubated for 20 min at 30 °C to allow cleavage of the nascent chain FLAG tag and expose the N-terminal glycine. Following 3C cleavage, purified NMT2 and NAC were diluted using cryo-EM buffer and were added to a final concentration of 1 and 0.5 μM, respectively. This mixture was incubated for another 20 min at 30 °C and moved immediately to ice prior to grid preparation. Quantifoil R2/1 300 mesh copper grids (Quantifoil, Q350CR1) were coated with 3.6 nm carbon and glow discharged at 15 mA for 15 s using an EMS glow discharger. Approximately 2 min prior to applying the sample to the grid, Myristoyl-Coenzyme A (Cayman Chemicals, #39777) was added into the sample to a final concentration of 100 μM. A 5 μL drop of the reaction was pipetted onto the grid and incubated for 1 min and 45 s in a 4 °C, 100% humidity-controlled chamber before plunge frozen in liquid ethane pre-cooled using liquid nitrogen. Data was collected on the ThermoFisher Krios, operated at 300 kV with a K3 direct electron detector at a total magnification of ×105,000 and an energy filter with a slit width of 10 eV. The defocus range used was from −1.8 to −0.4 μm with a step size of 0.2. A total of 16,267 exposures were taken and movies were collected at a pixel size of 0.83 Å and a total dose of 40e$^-$/Å$^2$ across 40 frames per movie.

## Cryo-EM data processing and refinement

Initial processing steps of motion-correction, dose-weighting, and contrast transfer function (CTF) estimation were completed in CryoSPARC (V4.6.0) (Punjani et al, 2017). Blob picker was used to select particles with a diameter range of 250–450 Å from the micrographs. Picks were inspected and a total of 902,088 particles were extracted from the micrographs at a box size of 512 × 512 pix and Fourier cropped to a box size of 256 × 256 pix for subsequent steps. The particles were subjected to 2D classification for a total of 150 classes, with 61 classes selected and the remaining 89 junk classes excluded. These particles underwent a round of ab initio refinement followed by a round of heterogeneous refinement. After excluding further junk particles and incomplete ribosomal particles, the remaining 424,885 particles were subjected to an initial homogenous refinement to yield a consensus map of the 80S ribosome with density for NAC and NMT2. These particles went through multiple rounds of 3D variability analysis and homogenous refinement following the initial consensus volume refinement. First, using a solvent mask of the entire volume, particles with no tRNA or small subunits that were not well resolved were filtered out. Then, using a mask of NAC and NMT2, particles were separated by the presence of NAC and NMT2 as well as different conformations of the proteins on the ribosome. The largest class was taken, re-extracted at a full 512 × 512 pix box size, and subjected to two rounds of 3D variability analysis with solvent masking and masking of the active site of NMT2. Ribosomes with strong P-site tRNA density and strong density of substrate within

**Table 1.** Cryo-EM data collection, refinement, and validation statistics.

|  | RNC_MARCKS:NMT2:NAC |
| --- | --- |
| *EMDB code* | EMD-49275 |
| *PDB code* | 9NDP |
| *Data collection and processing* |  |
| Nominal magnification | ×105,000 |
| Voltage (kV) | 300 |
| Electron exposure (e$^-$/Å$^2$) | 40 |
| Defocus range (μm) | −1.8/−0.4 |
| Pixel size (Å) | 0.83 |
| Initial particle images (no.) | 660,042 |
| Final particle images (no.) | 23,479 |
| Map resolution at FSC = 0.143 (Å) | 2.8 |
| *Structure refinement in PHENIX 1.20.1* |  |
| Model resolution at FSC = 0.5 (Å) | 3.0 |
| CC_mask | 0.85 |
| Map sharpening B factor (Å$^2$) | 34.0 |
| Model composition |  |
| Non-hydrogen atoms | 218,587 |
| Protein residues | 12,272 |
| RNA residues | 5585 |
| B factors min/max/mean (Å$^2$) |  |
| Protein | 7.14/165.25/67.36 |
| RNA | 0.00/232.40/74.96 |
| Ligand | 6.55/141.95/49.74 |
| RMSD |  |
| Bond lengths (Å) | 0.002 |
| Bond angles (°) | 0.427 |
| *Validation* |  |
| MolProbity score | 1.84 |
| Clashscore | 7.98 |
| Poor rotamers (%) | 2.15 |
| Ramachandran plot |  |
| Favored (%) | 97.17 |
| Allowed (%) | 2.80 |
| Outliers (%) | 0.03 |

the NMT2 active site were selected. The final 23,479 particles underwent a final round of homogenous refinement for a final global resolution of 2.82 Å as determined by the gold standard Fourier Shell Correlation (FSC) using two independently processed half-sets of particles using a threshold of 0.143. The local resolution of the map was also calculated using CryoSPARC (v4.6.0) with an FSC threshold of 0.5 (Table 1).

## Model building

After data processing, models of the 80S ribosome with Xbp1u-stalled nascent chain (PDB 6R5Q), NAC (PDB 7QWR), and

NMT2 (PDB 6PAU) were docked into the cryo-EM map with ChimeraX (v1.6.1). NMT2 and NAC were additionally fitted as rigid bodies into the map and adjusted in Coot (v0.9.8.93) (Emsley and Cowtan, 2004) using the observed densities. The NMT2 substrate chain was mutated to the properly registered MARCKS residues and fit into the observed sidechain density. The model underwent 14 macrocycles of real-space refinement to fit into the density map using PHENIX (v1.20.1) (Liebschner et al, 2019). The majority of clashes were corrected using secondary structure, sidechain rotamer, nucleotide, and Ramachandran restraints. The final model MolProbity scores were validated, and the model was subsequently used for figure generation in ChimeraX (v1.6.1) (Meng et al, 2023).

## Data availability

The cryo-EM map and the corresponding atomic model for the NMT2-NAC-RNC$_{MARCKS}$ structure have been deposited under the accession codes EMD-49275 and PDB-9NDP.

The source data of this paper are collected in the following database record: biostudies:S-SCDT-10_1038-S44318-025-00548-4.

## Peer review information

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

## Acknowledgements

We thank the Jomaa lab members for the helpful discussions. Cryo-EM data collection was conducted at the molecular electron microscopy core facility (RRID:SCR_019031) at the University of Virginia (UVA) School of Medicine, which was built with NIH grant G20-RR31199. We thank Yelena Peskova for making the NMT2 and NAC constructs for the mutational studies and Kevin Nguyen and Owen Albanese for help with protein purifications. We also thank Michael Purdy for assisting with the cryo-EM data collection and additional computational support. This work was supported by The Owens Family Foundation, by NIH grant 1R35GM160490, the Searle Scholars Program, Grant #: SSP-2023-106, and aided by Grant # 134088-IRG-19-143-33-IRG from the American Cancer Society to AJ. We acknowledge the support provided by the Molecular Biophysics Training Program at UVA to SZ through NIH T32GM158467-01 and the Cellular and Molecular Biology Training Program at UVA for support provided to EM through NIH T32GM139787-3.

## Author contributions

**Sara Zdancewicz**: Conceptualization; Resources; Data curation; Software; Formal analysis; Validation; Investigation; Visualization; Methodology; Writing—original draft; Project administration; Writing—review and editing. **Emir Maldosevic**: Investigation; Visualization; Methodology; Writing—review and editing. **Kinga Malezyna**: Methodology; Writing—review and editing. **Ahmad Jomaa**: Conceptualization; Data curation; Formal analysis; Supervision; Funding acquisition; Validation; Investigation; Visualization; Methodology; Writing—original draft; Project administration; Writing—review and editing.

Source data underlying figure panels in this paper may have individual authorship assigned. Where available, figure panel/source data authorship is listed in the following database record: biostudies:S-SCDT-10_1038-S44318-025-00548-4.

## Disclosure and competing interests statement

The authors declare no competing interests.

# Expanded View Figures

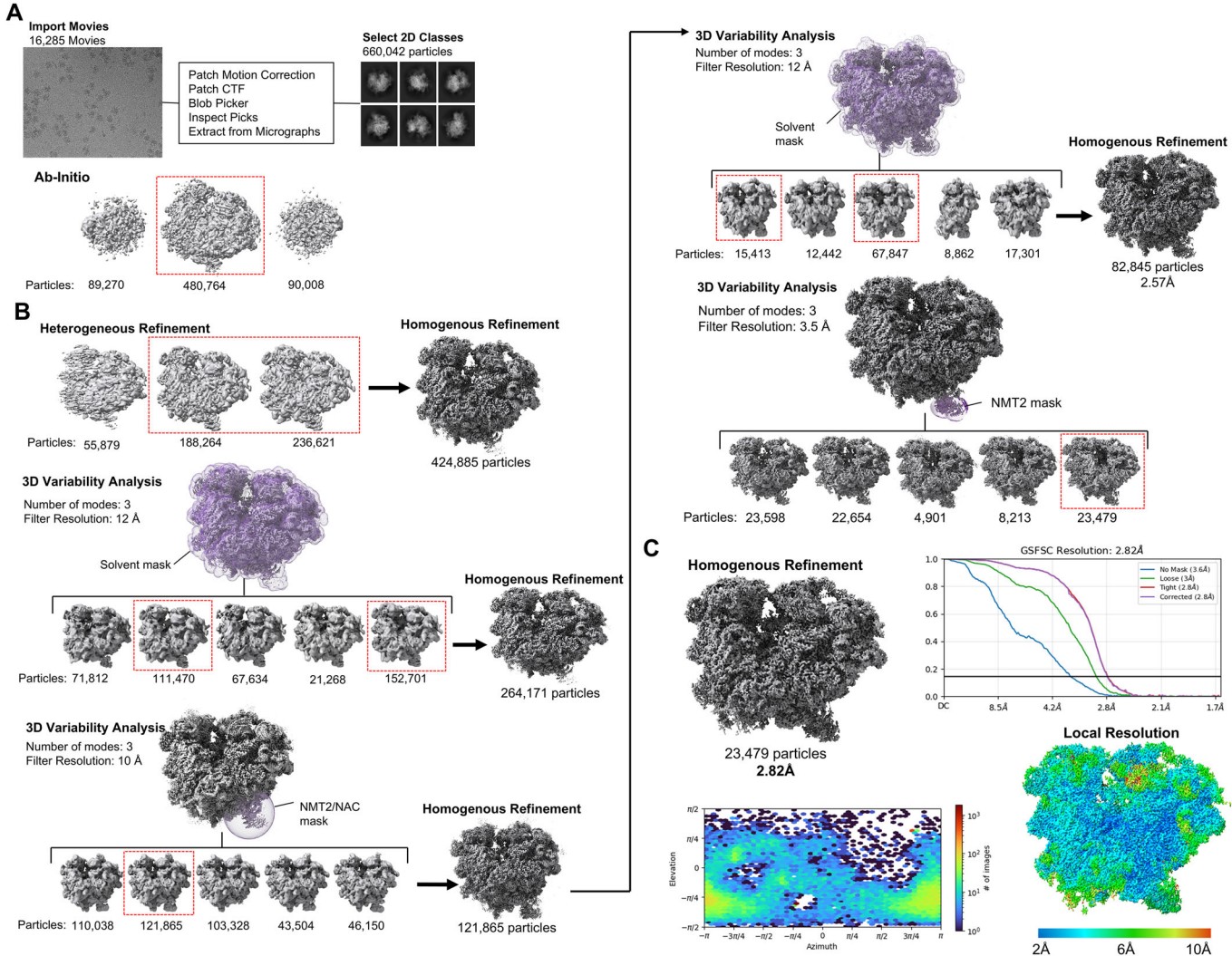

**Figure EV1. Cryo-EM processing scheme of RNC_MARCKS:NMT2:NAC ternary complex.**

(A) Movies initially collected were subjected to motion and CTF correction, and particles were identified for 2D classification using blob picker. 2D classes were selected and ab initio was used to separate out full ribosomal particles. (B) After initial heterogenous refinement to filter out bad classes, the 3D volumes underwent 4 subsequent rounds of 3D variability analysis using a solvent mask and NMT2/NAC mask sequentially. (C) The final volume was constructed from 23,479 particles with an overall resolution of 2.82 Å as determined by gold standard Fourier shell correlation of independent half maps with a cutoff of 0.143. Angular distribution plot and local resolution were determined in cryoSPARC (v4.6.0).

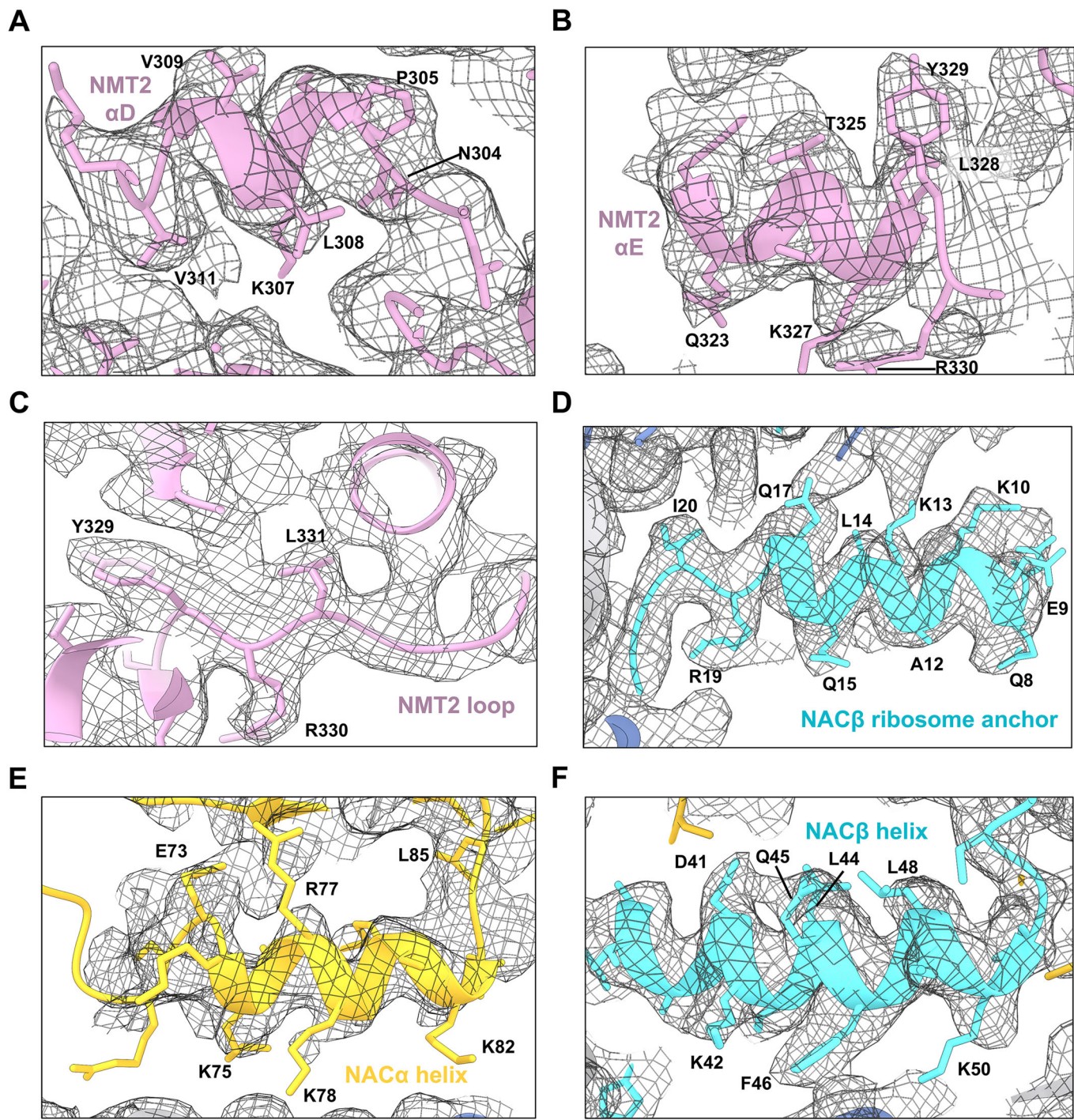

**Figure EV2. Structure of NMT2 and NAC within the ternary complex.**

(**A**) Cryo-EM map depicts fit of NMT2 αD into density. (**B**) NMT2 helix αE fit into cryo-EM density. (**C**) Cryo-EM map shows fit of the loop region connected to the anchor helix αE of NMT2 into density. (**D**) A close-up of the ribosome anchor of NACβ. (**E**) Close-up view of NACα helix within its ordered dimerization domain. (**F**) Helix within NACβ dimerization domain shown in a close-up view. All cryo-EM densities are shown as mesh and filtered to 3.5 Å.

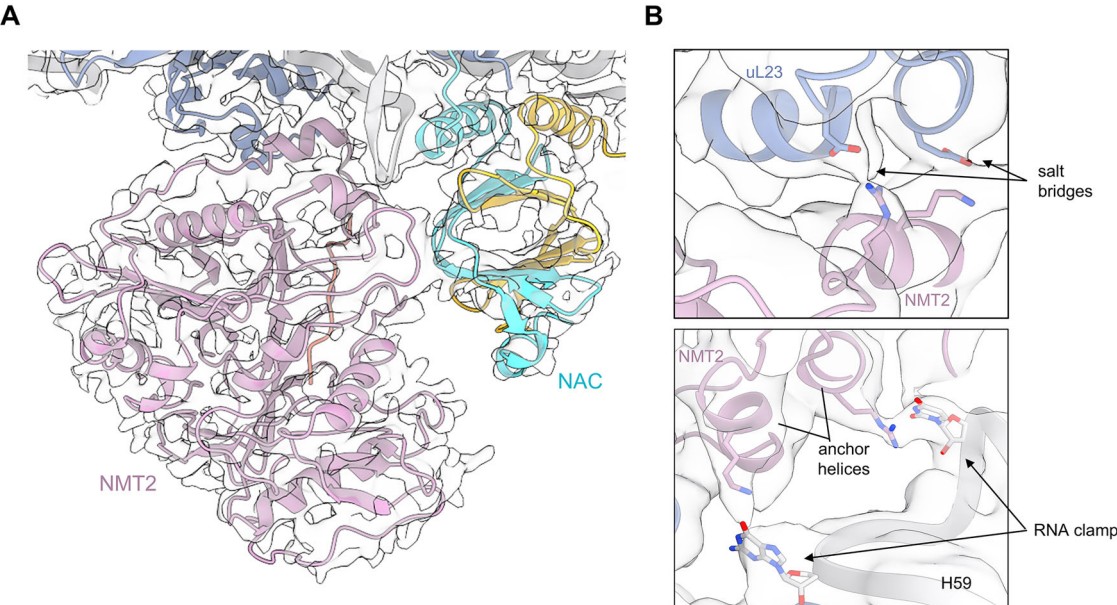

**Figure EV3. Fit of NMT2 and NAC models into the cryo-EM density map highlighting the indicated contacts.**

(A) Overall fit of NAC (cyan, gold) and NMT2 (pink) into cryo-EM density map shown as a transparent surface and filtered to 3.5 Å. (B) Close-up views of H59 clamp contacts and uL23 electrostatic contacts show density for both flipped-out bases as well as for interactions with NMT2.

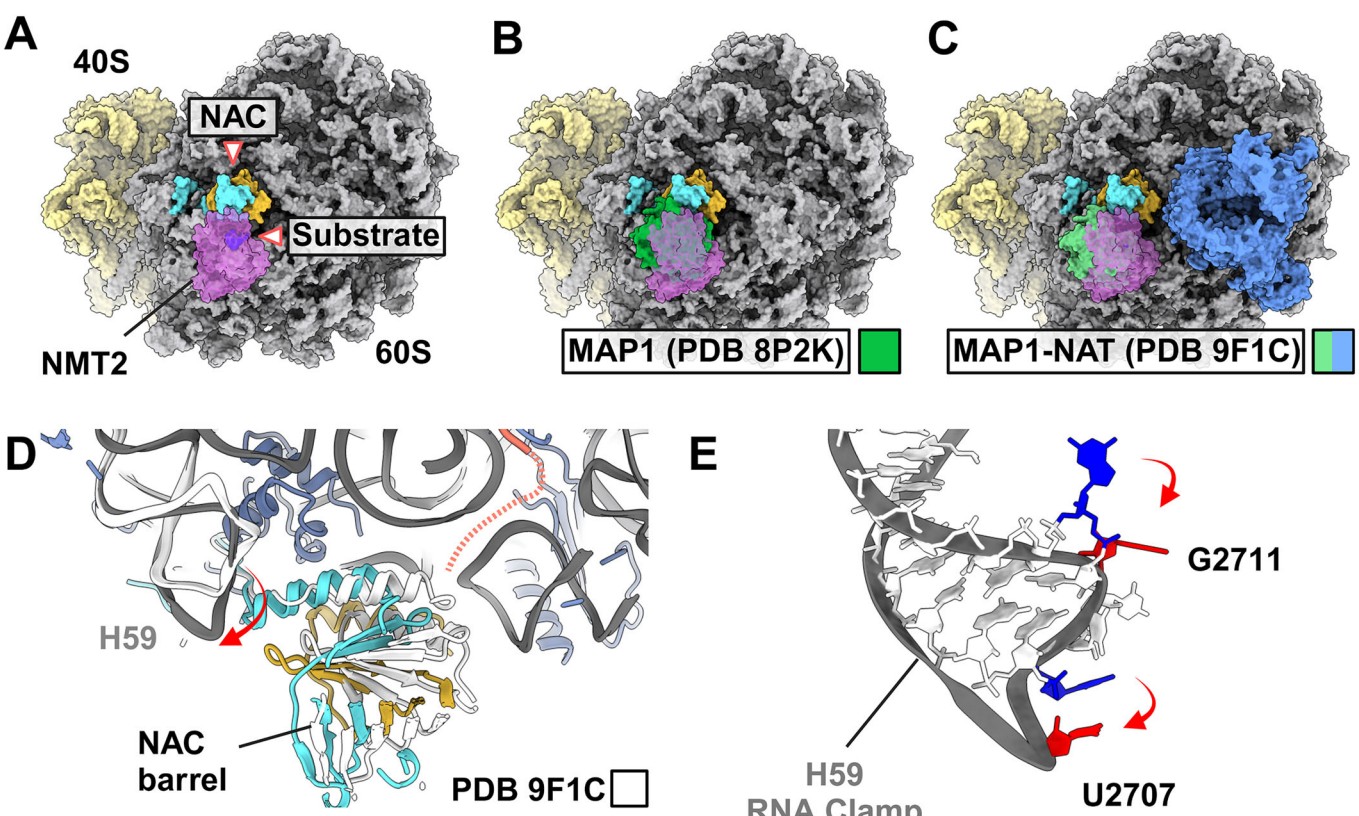

**Figure EV4.  MAP1 binding overlaps with NMT2 docking and substrate engagement on ribosomes.**

(**A**) Surface model of the current NAC-NMT2 structure with substrate bound in the active site (blue). (**B**) Model of MAP1 docked to the current structure showing overlap of MAP1 and NMT2 binding sites. (**C**) Model of MAP1-NAT docked to the current structure depicting differing binding locations for NAT compared to MAP1 and NMT2. (**D**) Conformational change of NAC and H59 (red arrow) between the current structure and the NAC-MAP1 (PDB 8P2K) and the NAC-MAP1-NAT structure (PDB 9F1C). (**E**) Rearrangement of bases of the RNA clamp compared to the NAC-MAP1-NAT structure (PDB 9F1C).

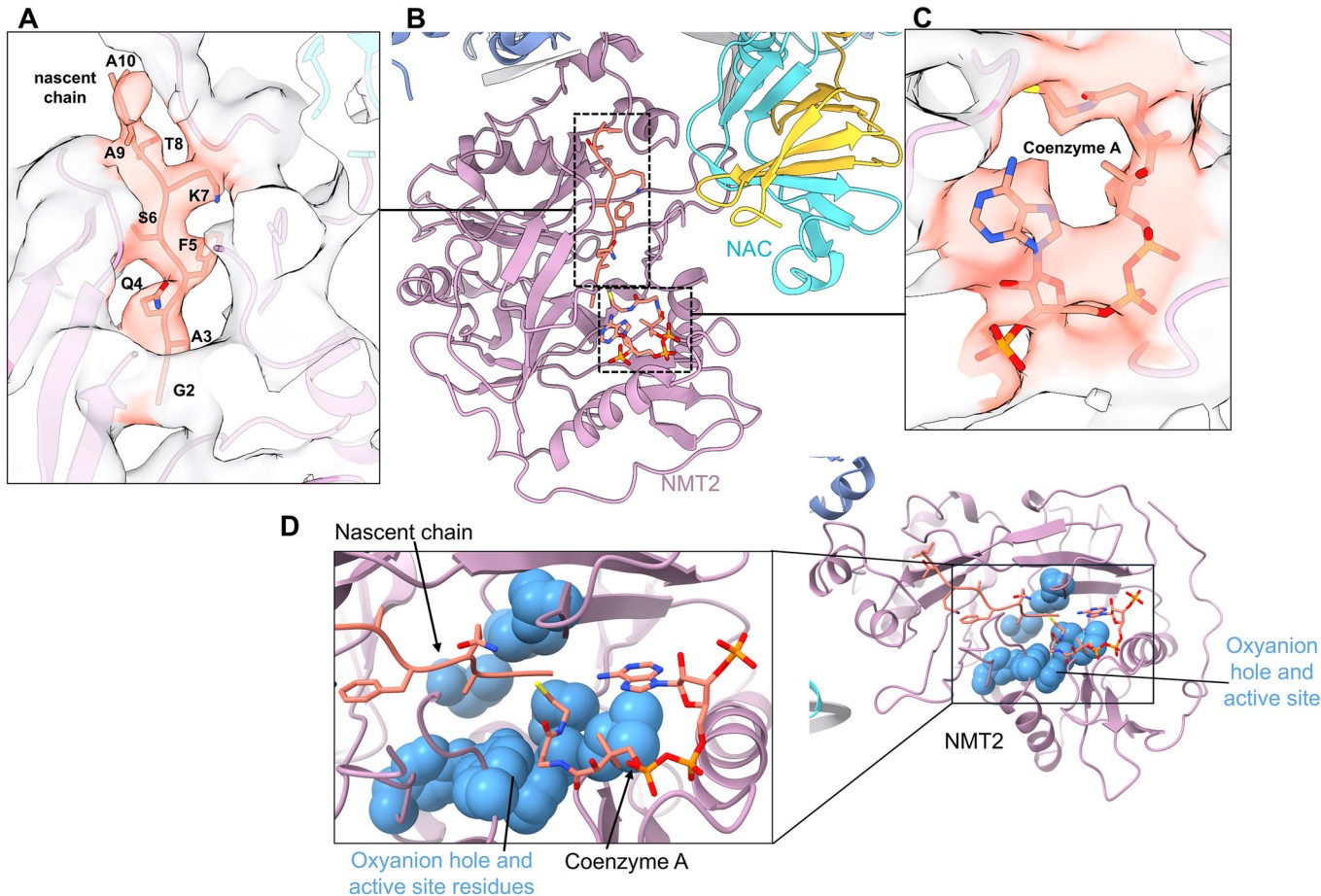

**Figure EV5. The nascent chain and coenzyme A are present in the RNC_MARCKS:NMT2:NAC ternary complex density.**

(A) N-terminus of MARCKS is present in cryo-EM density (coral). (B) Crystal structure of NMT2 (PDB ID 6PAU) shown docked onto the ribosome containing the substrate nascent chain and coenzyme A. (C) Density for coenzyme A present in cryo-EM density map (coral). (D) NMT2 showing the oxyanion hole and active site of the enzyme (blue) and where the nascent chain and coenzyme A are positioned during myristoylation. All cryo-EM density has been shown as a transparent surface and filtered to 3.5 Å.

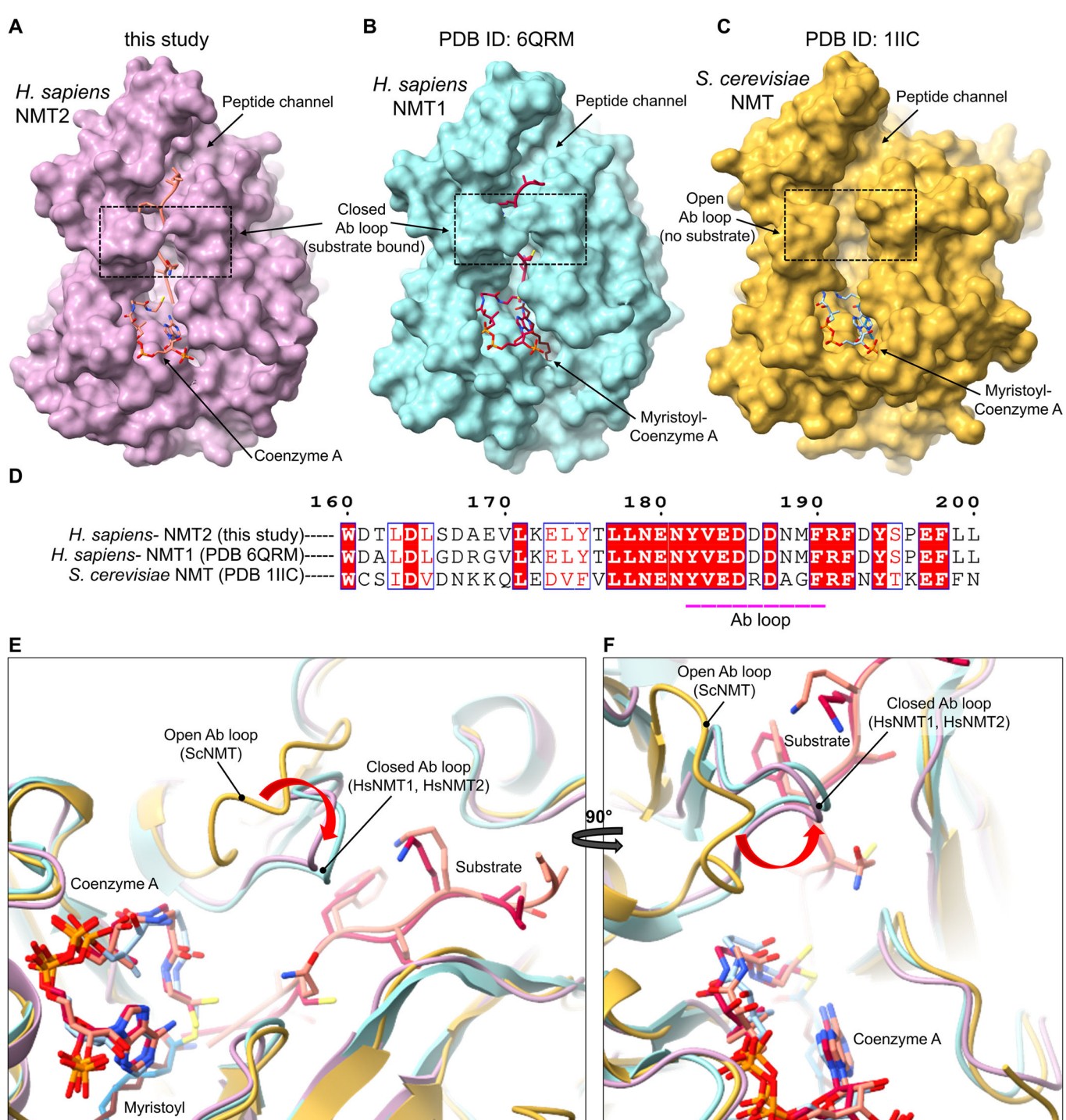

**Figure EV6.   Comparison of Ab loop conformation of NMT2 with peptide bound NMT1 and apo *S. cerevisiae* NMT.**

(**A**) Surface view of human NMT2 showing a closed Ab loop conformation over a substrate polypeptide (coral). Coenzyme A and the opening to the peptide channel where the substrate would enter from the ribosome polypeptide exit tunnel are also labeled. (**B**) Surface view of human NMT1 showing a closed Ab loop conformation over a substrate polypeptide (magenta). Myristoyl-coenzyme A and the opening to the peptide channel where the substrate would enter from the ribosome polypeptide exit tunnel are also labeled. (**C**) Surface view of *S. cerevisiae* NMT showing an open Ab loop conformation over an empty substrate channel. Myristoyl-coenzyme A (light blue) and the opening to the peptide channel are also labeled. (**D**) Sequence alignment of HsNMT2, HsNMT1, and ScNMT indicating the residues making up the Ab loop (magenta line). (**E**) Close-up view of the conformational change between open and closed Ab loops when a polypeptide substrate is present. NMTs are shown as cartoons (pink, blue, yellow) and overlaid. Substrate, myristoyl-coenzyme A, and coenzyme A are shown as sticks. (**F**) A second view of the open and closed Ab loop conformations in the various NMTs.

