## [Peer Review File · The EMBO Journal]

NAC Couples Protein Synthesis with Nascent Polypeptide Myristoylation on the Ribosome

Sara Zdancewicz, Emir Maldosevic, Kinga Malezyna, and Ahmad Jomaa

Corresponding author(s): Ahmad Jomaa (ahmadjomaa@virginia.edu)

Review Timeline:

Submission Date:	28th Feb 25
Editorial Decision:	17th Apr 25
Revision Received:	4th Jun 25
Editorial Decision:	14th Jul 25
Revision Received:	15th Jul 25
Accepted:	23rd Jul 25

Editor: *Cornelius Schneider*

Transaction Report:

Dear Prof. Jomaa,

Thank you for submitting your manuscript for consideration by the EMBO Journal. It has now been seen by three referees whose comments are shown below.

As you can see all three referees think that the results are interesting and important. However, referees #2 and especially referee #3 also raise a number of major concerns which I think are reasonable and productive.

Given the referees' positive recommendations, I would like to invite you to submit a revised version of the manuscript, addressing the comments of all three reviewers. I should add that it is EMBO Journal policy to allow only a single round of revision, and acceptance of your manuscript will therefore depend on the completeness of your responses in this revised version.

Thank you for the opportunity to consider your work for publication. I look forward to your revision.

Yours sincerely,

Cornelius Schneider, PhD
Editor
The EMBO Journal
c.schneider@embojournal.org

We realize that it is difficult to revise to a specific deadline. In the interest of protecting the conceptual advance provided by the work, we recommend a revision within 3 months (15th Jul 2025). Please discuss the revision progress ahead of this time with the editor if you require more time to complete the revisions. Use the link below to submit your revision:

Referee #1:

Review of Zdancewicz et al. "NAC couples protein synthesis with nascent polypeptide myristoylation on the ribosome"
The manuscript of Zdancewicz et al presents a lovely structure of the myristoylation enzyme NMT2 bound to a translating ribosome. Co-translational modification of proteins is an important control for spatial targeting of enzymes, and myristoylation leads to membrane localization. NMT2 has been shown to bind to translating ribosomes and to modify an N-terminal glycine residue of the appropriate sequence context after cleavage of the N-terminal methionine. Here the authors capture a translating ribosome on a substrate for myristoylation, testing different length sequences to present a so-called MARCKS sequence outside the ribosomal peptide tunnel. Their biochemical results show that NMT2 and the general nascent chain factor NAC co-sediment in the 80S fraction, and the authors then isolate this complex and solve a cryoEM structure at good resolution. The structure shows that NMT2 and NAC interact at the peptide exit tunnel essentially extending the tunnel by an addition 44Å, with the N-terminus of the MARCKS peptide showing clear density in the active site of the enzyme. The authors then test aspects of the structure biochemically, in particular the role of observed NMT2-ribosome interactions, and NAC-NMT2 interactions by mutagenesis of key residues in either NMT2 or NAC. These results confirm the observed importance of the protein-protein and protein-ribosome interactions for formation of this complex. The results are crisply and clearly presented in the manuscript, while the figures themselves are also clear. In summary, this is an important piece of structural work that adds to our understanding of co-translational protein processing, and deserves publication in EMBO J. There are several minor points that should be addressed by the authors, and maybe an experiment or two.

1. The authors should discuss more deeply the nascent peptide substrate specificity for NMT2 and how their structure addresses this (based on the nice density in the active site).
2. A major conclusion of this work is that NAC and NMT2 form a protective tunnel for the nascent chain outside the ribosome, funneling the substrate N-terminus into the active site of NMT2. Do the authors believe that the intervening peptide sequence outside the ribosome (for which no density is observed) is within this tunnel or outside. Is there room or must there be compaction? The authors should try protease sensitivity of labeled versions of the stalled complex to assess accessibility.

Referee #2:

The manuscript EMBOJ-2025-120636 by Zdancewicz et al., shows structural insight into ribosome and nascent polypeptide-associated complex (NAC) recruitment of NMT2 for N-glycine myristoylation. The authors show by biochemical analyses that NMT2 interacts with ribosomes and that this interaction is stimulated by presence of NAC. Using MARCKS protein as a construct authors show cryoEM structure of ribosome complex with NMT2, NAC and nascent polypeptide chain. Authors make series of mutations and deletions to propose the model where NAC β recruits NMT2 to ribosome and NAC then orients and stabilize NAC-NMT2 complex on ribosome for myristoylation reaction. The structure of the ribosome-NAC-NMT2 complex is novel and of significance for the field of translation and membrane associated proteins. However there is one major and couple of minor concerns that authors need to address prior to publication.

Major concern

Based on the model authors argue that NAC β recruits NMT2 to ribosome, however in multiple figures NMT2 is recruited to ribosomes even in the absence of NAC. The input amounts (%) of NMT2 are hardly visible in almost all of the figures (not sure how they correspond to bound fraction) while binding to ribosomes is clearly visible. As such, it seems also plausible that NMT2 binds to ribosome without NAC, and if anything NAC as well as nascent polypeptide chain could stabilize its interaction with ribosomes. It is not clear based on currently presented data why authors are biased to proposed model.

Additional minor issues:

Line 97-98

While it could be problem with sensitivity and images, I am not sure that NMT2 is detected in all ribosomal fractions. It's

association with ribosomes is mostly with subunits and monosome fraction (fractions 3- 7, Figure 1B). I am not sure why either NMT2 or NAC should be located in fractions with 40S subunit.

Line 118-119

I am not sure if reported mild increase in NMT2 binding to ribosomes with chains longer than 55 is significant. As a matter of fact one would like to compare (as an additional control) binding of NMT2 to ribosomes which are incubated with harringtonine (translation initiation inhibitor) prior to adding RNA for MARCKS construct. This would indicate background level of NMT2 binding to ribosomes or subunits.

Lines 142-160

Fig 3A is not cited in the text at all.

Lines 176-185

It is hard to estimate binding in Fig 4 given that input for NMT2 WT is not really visible. What % of input was loaded? Why input for mutants was not shown?

Are NMT2 double and triple mutant (R324E and K327E, + R306E) expected to show reduced levels of binding in the presence of NAC? 2.5 fold stimulation of binding by NAC is being lost in these samples. Are these proteins having additional structural changes that impact interactions with NAC?

Referee #3:

Review

The manuscript by Zdancewicz et al. investigates the mechanism behind N-glycine myristoylation, a vital protein modification linked to protein synthesis on ribosomes, focusing on the role of N-myristoyltransferase 2 (NMT2) and the nascent polypeptide-associated complex (NAC). The researchers demonstrate that NAC enhances the binding of NMT2 to translating ribosomes, forming a complex that governs the transfer of a myristoyl group to nascent peptides as they emerge from the ribosome's exit tunnel. Structural analysis using cryo-electron microscopy uncovers a continuous channel that guides the nascent chain to the catalytic site of NMT2, revealing the molecular details of how NAC coordinates this process. The findings highlight NAC as a crucial regulator of protein biogenesis, suggesting that its C-terminal tail recruits NMT2 to the ribosome and orients it for effective engagement with nascent substrates.

This research enhances the understanding of co-translational modifications, emphasizing their significance in maintaining proper protein functionality and cellular processes. This is a valuable manuscript, which presents important results and provides consequential insights into the mechanisms of co-translational myristoylation that are important to a broad range of experts in biomedical research. That said, the manuscript is not without shortcomings that should be addressed prior to acceptance, including:

Major points:

1. The biggest issue of the manuscript is data presented in Figure 5 which seems to have suffered from a lack of time before submission. The MSA demonstrating that "contact points are conserved in higher eukaryotes" only shows four species other than human. The co-sedimentation assays should be done consistently in triplicates and quantified after Figure 4. However, the most striking is the statement "The truncations of the flexible tails did not impair NAC binding to ribosomes which is mediated by the N-terminal anchor of NAC β (Fig 5D)" which refers to no data but a schematic drawing. Moreover, the data presented in Figure 5F seem to directly contradict this statement and all the following interpretations based on this experiment, as the co-sedimentation is showing no signal for any of the truncated versions of NAC α or NAC β . While this can be caused by antibody reactivity (for both N- and C- terminal truncations though), the authors need to show that these NAC truncation variants are still a part of the tested complexes (and remain soluble), otherwise all interpretations stemming from these data in Results and Discussion are invalid.
2. In Figure 4G and H and 5F and G (when quantified), the authors should be normalizing the quantification to a load control (RPL10a or other control), since ribosome load difference would affect the co-sedimented NMT2 signal as well. There seem to be load differences judging by the RPL10a signal intensities and the normalization is unclear as it is described differently in Methods and figure legends. Importantly, the authors should present the results for the input samples of the co-sedimentation assays, which are never shown, even though they were analyzed according to the Methods.
3. The part where the authors claim: "The N-terminal tail of NMT2 contains an unstructured region that has previously been shown necessary for ribosomal localization.²⁵ To investigate whether this region is also involved in NAC binding and subsequent recruitment to the ribosome, we generated an NMT2 Δ N mutant lacking the first 105 amino acids." raises important questions. How does the NMT2 Δ N mutant test for NAC binding specifically rather than for general association with the complex (i.e. via other interactions)? The claim of subsequent recruitment is also not supported by data and deserves a separate point.
4. The authors claim to test that "this region is also involved in NAC binding and subsequent recruitment to the ribosome" and that "First, the C-terminal tail of NAC β recruits NMT2 to the ribosome. NAC then orients NMT2 on the ribosome, with the NAC-NMT2 loop-barrel contact forming to stabilize the ternary complex." While presenting no data proving such sequential order of binding events. The authors need to either clearly present such data or tone down their statements to avoid this discrepancy.
5. The authors discuss that "NAC and NMT2 extends the polypeptide exit tunnel, generating a protected channel to the catalytic core of NMT2" and that "the protective channel formed by the ternary complex may be beneficial during co-translational nascent chain processing to shield the polypeptide". At the same time, much longer nascent chains were associated with NMT2 with same apparent efficiency (Fig 1E, F) and in the structure this "channel" seems to be at least half open (as presented in the scheme of Figure 6). The authors should reconcile this with their idea of the protective channel. In this sense, Figure 3 is quite

misleading. Is there still room for the longer nascent chains to loop out of the tunnel as the structure and biochemical data suggest? Can the authors show that these interactions with longer nascent chains are still productive? It would greatly benefit the study to show the myristoylation activity directly for the reconstitution approach used for the structural part. Methods utilizing antibodies against myristoyl-glycine seem to be particularly useful here.

6. The molecular model has several issues that have to be resolved: Chains s, t, 0 and RR are not supported by experimental data (density) and should not be included in the model. Model quality is generally poor in some aspects like the clashscore and especially the sidechain outliers, which would put this structure in the worst 6.4% in this regard. This is quite striking since the authors claim that "Any clashes were corrected using secondary structure, sidechain rotamer, nucleotide, and Ramachandran restraints." In the Methods.

Minor points:

1. Both the NMT2 and NAC seem to be enriched in the 40S fraction in Fig 1B. How do the authors explain this? The authors should also include fractions 1 and 2 (which are not shown) so that the gradient profile is aligned with the WB fractions and the unbound fractions can be seen.
2. The proposed histidine stacking interaction presented in Figure 5A is not clearly visible and was not observed in the model the authors have provided. For any claims on the importance of these residues in the interaction, the authors should present more convincing mutation analysis (e.g., mutating both H411 and H412 at the same time).
3. The authors write about "the ternary complex" in the Abstract, which has not been introduced. It would be better to specify the complex, especially since the term ternary complex has a very prominent established meaning in the translation field.
4. Figures 1A, 3A, Extended Data Figures 1, 6A-B and E, are not mentioned in the manuscript and sometimes latter panels are referred to before the initial ones (panel 4B for instance). The authors should fix these issues.
5. The authors should specify and show how they "engineered a 3C cleavage site into the sequence" of NMT2. Was it necessary to mutate the second NMT2 residue to Pro to achieve the optimal the 3C protease cleavage motif?
6. There is no "partial density" apparent for the myristoyl group in Extended Data Figure 6D. The authors should either clearly present such density or tone down their statement.
7. There are no data presented for the statement "This anchor helix contains several positively charged residues that establish electrostatic interactions with uL23."
8. There seems to be no reaction for the "added to the reaction and pipetted gently" on line 394.
9. There is nothing regarding the in vitro translation described above for the claim "crude RNCs were generated by reacting mRNA generated by in vitro transcription with RRLs (Promega, L4540) as described above". Even later, for the cryo-EM sample preparation, this should be described more clearly for non-expert readers.
10. Why was "pipetting up and down gently to avoid bubbles" used with the SDS sample buffer to prepare SDS PAGE samples? (Methods, line 436)
11. What does it mean that Myristoyl-Coenzyme A "was spiked in"? (Methods, line 494)
12. The Coenzyme A depicted in Fig 2C is presented as stick model, not cartoon as specified in the legend.
13. Quantification of band intensities should be added to support the claims regarding data in Figure 1F and Figure 5B.
14. Lines 202-203 - Sentence about the sequence alignment of the NMTs in the loop-barrel interaction region with the corresponding link to Figure 5C is missing and this panel is only referred to at the end of Discussion.
15. Lines 282-283 - It would be fair to mention that the observed interaction between NMT2 and NAC β , that is hypothesized to stabilize NMT2 in favorable conformation relative to the nascent chain, is not universally conserved, as the *A. thaliana* NMT2 is clearly missing both histidine residues, as depicted in the alignment in Figure 5C.
16. The authors should check and correct the units they are using in the Methods section (RPM instead of g throughout Methods, concentration of A260, e/ \AA^2).
17. Lines 402, 420, 435 - Rotor used for the ultracentrifugation is not specified.
18. Software versions used in the study should be specified in the Methods.
19. Line 684 - Year of the cited article is missing.

Point-by-point answer to reviewers

Reviewer #1

The manuscript of Zdancewicz et al presents a lovely structure of the myristoylation enzyme NMT2 bound to a translating ribosome. Co-translational modification of proteins is an important control for spatial targeting of enzymes, and myristoylation leads to membrane localization. NMT2 has been shown to bind to translating ribosomes and to modify an N-terminal glycine residue of the appropriate sequence context after cleavage of the N-terminal methionine. Here the authors capture a translating ribosome on a substrate for myristoylation, testing different length sequences to present a so-called MARCKS sequence outside the ribosomal peptide tunnel. Their biochemical results show that NMT2 and the general nascent chain factor NAC co-sediment in the 80S fraction, and the authors then isolate this complex and solve a cryoEM structure at good resolution. The structure shows that NMT2 and NAC interact at the peptide exit tunnel essentially extending the tunnel by an addition 44Å, with the N-terminus of the MARCKS peptide showing clear density in the active site of the enzyme. The authors then test aspects of the structure biochemically, in particular the role of observed NMT2-ribosome interactions, and NAC-NMT2 interactions by mutagenesis of key residues in either NMT2 or NAC. These results confirm the observed importance of the protein-protein and protein-ribosome interactions for formation of this complex. The results are crisply and cleanly presented in the manuscript, while the figures themselves are also clear. In summary, this is an important piece of structural work that adds to our understanding of co-translational protein processing, and deserves publication in EMBO J. There are several minor points that should be addressed by the authors, and maybe an experiment or two.

Minor concerns:

1. The authors should discuss more deeply the nascent peptide substrate specificity for NMT2 and how their structure addresses this (based on the nice density in the active site).

We thank the reviewer for the feedback and supportive evaluation of our work. We are especially grateful for the recognition of the significance of our structural and biochemical findings.

We have expanded the Results to provide more detail on the interaction between the nascent chain and the catalytic site of NMT2 (lines 157-168). This is now supported by an additional panel in Figure 2 (panel E) and a new supplementary figure (Figure EV6), which compares our structure with available X-ray structures of *H. sapiens* NMT1 bound to a peptide substrate, and with *S. cerevisiae* apo NMT.

2. A major conclusion of this work is that NAC and NMT2 form a protective tunnel for the nascent chain outside the ribosome, funneling the substrate N-terminus into the active site of NMT2. Do the authors believe that the intervening peptide sequence outside the ribosome (for which no density is observed) is within this tunnel or outside. Is there room or must there be compaction? The authors should try protease sensitivity of labeled versions of the stalled complex to assess accessibility.

We appreciate this thoughtful suggestion and have decided to test it experimentally. To assess whether the nascent chain segment between the ribosome and NMT2 is protected, we generated a new construct containing a TEV protease cleavage site in the middle of the nascent chain. We tested those on the 74-amino-acid construct used in our cryo-EM studies and compared TEV sensitivity in the presence and absence of NMT2. Our results show a modest level of protection upon NMT2 addition, suggesting some shielding of the nascent chain. However, the effect was not robust enough to confidently conclude that the NAC-NMT2 arrangement forms a functionally protective channel.

Therefore, we opted not to include this dataset in the main manuscript and have toned down the text to present a more conservative interpretation of the channel's role. For transparency, we present the results of the reviewer's proposed experiment below.

Reviewer #2-

The manuscript EMBOJ-2025-120636 by Zdancewicz et al., shows structural insight into ribosome and nascent polypeptide-associated complex (NAC) recruitment of NMT2 for N-glycine myristoylation. The authors show by biochemical analyses that NMT2 interacts with ribosomes and that this interaction is stimulated by presence of NAC. Using MARCKS protein as a construct authors show cryoEM structure of ribosome complex with NMT2, NAC and nascent polypeptide chain. Authors make series of mutations and deletions to propose the model where NAC β recruits NMT2 to ribosome and NAC then orients and stabilize NAC-NMT2 complex on ribosome for myristoylation reaction. The structure of the ribosome-NAC-NMT2 complex is novel and of significance for the field of translation and membrane associated proteins. However there is one major and couple of minor concerns that authors need to address prior to publication.

1. Based on the model authors argue that NAC β recruits NMT2 to ribosome, however in multiple figures NMT2 is recruited to ribosomes even in the absence of NAC. The input

amounts (%) of NMT2 are hardly visible in almost all of the figures (not sure how they correspond to bound fraction) while binding to ribosomes is clearly visible. As such, it seems also plausible that NMT2 binds to ribosome without NAC, and if anything NAC as well as nascent polypeptide chain could stabilize its interaction with ribosomes. It is not clear based on currently presented data why authors are biased to proposed model.

We very much thank the reviewer for their thoughtful comments and for highlighting the significance of our work. We appreciate the opportunity to clarify our proposed model regarding NAC β -mediated recruitment of NMT2 to the ribosome.

Our model that NAC β recruits NMT2 to the ribosome is supported by several lines of reasoning:

1. **Temporal and positional context:** Prior studies have established that NAC is constitutively associated with translating ribosomes (PMID: 21765803; 17229726; 17229726)
2. Given the ubiquitous presence of NAC on the ribosome and the low expression of NMT2 in cells, it is likely that NAC occupies the ribosome exit site prior to the engagement of NMT2, placing it in a position to facilitate NMT2 recruitment.
3. **Precedent from analogous systems:** NAC has been shown to promote ribosome association of other nascent chain interacting factors such as SRP, MAP1, and Nata (PMID: 17229726; 17229726; 17229726)
4. Although these factors can also bind ribosomes independently in vitro, NAC enhances their recruitment and/or stabilization in a cellular context. This precedent supports a similar mechanism for NMT2 and is in line with our results.
5. **In vitro binding of NMT2 in the absence of NAC:** We agree with the reviewer that NMT2 can associate with ribosomes in the absence of NAC in vitro, as seen in some of our experiments. However, this observation does not contradict our model. As with SRP for example, in vitro assays may not fully capture the regulatory or hierarchical nature of factor recruitment observed in vivo.

However, we also acknowledge that NAC may contribute both to the recruitment and stabilization of NMT2 on the ribosome. Our structural data support additional interaction points between NMT2 and NAC, particularly through the β -subunit and its C-terminal tail, suggesting that NAC not only facilitates initial binding but also stabilizes the complex for efficient myristoylation.

Therefore, we revised the manuscript both in abstract and discussion (highlighted text on page 12-13). The schematic shown in Figure 1A and Figure 7 were also modified to more explicitly state that NAC may contribute to both recruitment and stabilization of NMT2 and clarify the interpretation of our in vitro binding data accordingly.

Reviewer #2- Minor comments:

1. Line 97-98- While it could be problem with sensitivity and images, I am not sure that NMT2 is detected in all ribosomal fractions. It's association with ribosomes is mostly with subunits and monosome fraction (fractions 3- 7, Figure 1B). I am not sure why either NMT2 or NAC should be located in fractions with 40S subunit.

We acknowledge that the sensitivity in our initial profile was low, likely due to suboptimal treatment with cycloheximide, which may have resulted in incomplete stalling of translating ribosomes and nascent chains. We have repeated the experiment with optimized cycloheximide treatment which was added during the culturing of the cells. This treatment improved the quality of the gradient profile (new Figure 1, panel B).

In the revised data, we now observe more robust association of both NMT2 and NAC with ribosomal subunits, monosomes, and polysomes. However, we consistently detect a portion of free protein at the top of the gradient, particularly for NAC, which trails into the 40S fraction. We speculate that this may represent unbound protein present in the cytoplasm or protein that dissociates during the gradient centrifugation step.

2. Line 118-119- I am not sure if reported mild increase in NMT2 binding to ribosomes with chains longer than 55 is significant. As a matter of fact one would like to compare (as an additional control) binding of NMT2 to ribosomes which are incubated with harringtonine (translation initiation inhibitor) prior to adding RNA for MARCKS construct. This would indicate background level of NMT2 binding to ribosomes or subunits.

Based on reviewer's suggestion, we performed an additional control using salt-washed and puromycin-treated HEK293 ribosomes, which effectively remove associated factors, mRNA, tRNAs, and nascent polypeptides. As shown in the Western blot (now included in Figure 1F), NMT2 binding to these stripped ribosomes is similar to ribosome containing nascent chains.

In light of this observation and the reviewer's comment, we have modified the sentence describing a mild increase in NMT2 binding to ribosomes with longer nascent chains.

The new statement on page 6 read as follows: “ *Our results indicate that NMT2 interacts with RNC_{MARCKS} and empty ribosome similarly with modest increase to RNC_{MARCKS} displaying 74 amino acids (Fig 1F).*” Line 119

3. Lines 142-160- Fig 3A is not cited in the text at all.

A citation for this figure has been added.

4. Lines 176-185- It is hard to estimate binding in Fig 4 given that input for NMT2 WT is not really visible. What % of input was loaded? Why input for mutants was not shown?

We have added immunoblots of the input samples used in the co-sedimentation assays to the supplementary figures to confirm that all components were present at similar levels. We also now include three independent replicates of the binding experiments, along with quantification of the bands to compare NMT2 WT and mutants.

The gels have been added in Main Figures 4, 5, and 6.

Because of differences in protein concentration between input and pellet fractions, and variability in antibody signals across blots, calculating % input is not reliable. The input blots are provided for qualitative validation only and were not used for quantitative analysis.

5. Are NMT2 double and triple mutant (R324E and K327E, + R306E) expected to show reduced levels of binding in the presence of NAC? 2.5 fold stimulation of binding by NAC is being lost in these samples. Are these proteins having additional structural changes that impact interactions with NAC?

We thank the reviewer for this insightful comment. To assess whether NAC stimulation is affected in the NMT2 mutants, we re-analyzed the data by plotting the fold change in ribosome binding in the presence of NAC (see below left panel). This analysis shows that the single (R306E) mutant exhibits similar NAC-dependent stimulation compared to wild-type NMT2, despite overall reduced binding. In contrast, the double (R324E/K327E) and triple (R306E/R324E/K327E) mutants show a clear loss of NAC-mediated enhancement.

R/K to E mutations are charge-reversal substitutions that retain similar side-chain size. In our constructs, these residues are surface-exposed and unlikely to affect the overall protein fold. To evaluate potential structural effects, we used AlphaFold 3 to model the mutants, and no major structural changes were predicted with RMSD of 1.0 Å compared to WT (attached below right panel). We therefore interpret the reduced binding primarily as impaired interaction with the ribosome, rather than disrupted binding to NAC.

(left panel) Quantification of how much NMT2 band intensity changes in the presence of WT NAC compared to NMT2 alone. This is completed for R306E, R324E/K327E, and R306E/R324E/K327E as well. The double and triple mutants show significantly decreased difference in ribosomal cosedimentation (p-values =0.0379 and 0.0153, respectively).

(right panel) AlphaFold 3 modeling of the triple mutant overlaid with the WT NMT2 structure shows no noticeable secondary structure changes between the two constructs. The triple mutant is shown since it has the most severe change from WT NMT2 out of the three mutants being discussed.

Reviewer #3-

The manuscript by Zdancewicz et al. investigates the mechanism behind N-glycine myristoylation, a vital protein modification linked to protein synthesis on ribosomes, focusing on the role of N-myristoyltransferase 2 (NMT2) and the nascent polypeptide-associated complex (NAC). The researchers demonstrate that NAC enhances the binding of NMT2 to translating ribosomes, forming a complex that governs the transfer of a myristoyl group to nascent peptides as they emerge from the ribosome's exit tunnel. Structural analysis using cryo-electron microscopy uncovers a continuous channel that guides the nascent chain to the catalytic site of NMT2, revealing the molecular details of how NAC coordinates this process. The findings highlight NAC as a crucial regulator of protein biogenesis, suggesting that its C-terminal tail recruits NMT2 to the ribosome and orients it for effective engagement with nascent substrates. This research enhances the understanding of co-translational modifications, emphasizing their significance in maintaining proper protein functionality and cellular processes. This is a valuable manuscript, which presents important results and provides consequential insights into the mechanisms of co-translational myristoylation that are important to a broad range of experts in biomedical research. That said, the manuscript is not without shortcomings that should be addressed prior to acceptance.

We thank the reviewer for their thorough and constructive assessment of our manuscript. We are pleased that the reviewer recognizes the significance of our findings and their broader relevance to the field of co-translational protein modifications.

Major comments:

1. The biggest issue of the manuscript is data presented in Figure 5 which seems to have suffered from a lack of time before submission. The MSA demonstrating that "contact points are conserved in higher eukaryotes" only shows four species other than human. The co-sedimentation assays should be done consistently in triplicates and quantified after Figure 4. However, the most striking is the statement "The truncations of the flexible tails did not impair NAC binding to ribosomes which is mediated by the N-terminal anchor of NAC β (Fig 5D)" which refers to no data but a schematic drawing. Moreover, the data presented in Figure 5F seem to directly contradict this statement and all the following interpretations based on this experiment, as the co-sedimentation is showing no signal for any of the truncated versions of NAC α or NAC β . While this can be caused by antibody reactivity (for both N- and C- terminal truncations though), the authors need to show that these NAC truncation variants are still a part of the tested complexes (and remain soluble), otherwise all interpretations stemming from these data in Results and Discussion are invalid.

We appreciate the reviewers for their suggestion to improve the interpretations for the experiments done in Figure. In response, we have addressed each concern raised:

1. **Expanded Sequence Alignment:** The multiple sequence alignment has been updated to include a broader range of species beyond the original four, supporting the conservation of contact residues in higher eukaryotes.

2. **Replication and Quantification:** All co-sedimentation assays in the original Figure 5 (now Figures 5 and 6) have been repeated in triplicate. Quantification has been added, and input gels for each experiment are now provided to support interpretation in the supplementary material and methods.
3. **Clarification on NAC Truncation Data:** While we agree with the reviewer that NAC tail truncations may affect antibody reactivity, we note that the interpretation that Figure 5F do not contradict our conclusions. Depending on the specific tail truncation, either NAC α or NAC β retains detectable binding to ribosomes as shown in the gel blotted for NAC α and NAC β and form NAC $\alpha\beta$ as validated during purification(see point below). We have clarified this in the text and now include the relevant blots in the revised figure.
4. **Validation of Complex Formation and Solubility:** To ensure that the observed effects are not due to loss of NAC $\alpha\beta$ complex formation or insolubility of the mutants, we tested all NAC truncation variants for their ability to assemble into heterodimers. A new SDS gel confirming the purified NAC complex and solubility of the NMT2 truncation mutant (NMT2 Δ N) has been added to the revised figure set. This gel has been added to Appendix Figure S4 (panel A).

2. In Figure 4G and H and 5F and G (when quantified), the authors should be normalizing the quantification to a load control (RPL10a or other control), since ribosome load difference would affect the co-sedimented NMT2 signal as well. There seem to be load differences judging by the RPL10a signal intensities and the normalization is unclear as it is described differently in Methods and figure legends. Importantly, the authors should present the results for the input samples of the co-sedimentation assays, which are never shown, even though they were analyzed according to the Methods.

All quantifications were normalized to the loading control RPL10a. Figure legends for Figures 1, 4H, and 4I have been updated to specify this more clearly

“weighting by RPL10a band intensity per lane,” consistent with the gel quantification described in the Methods section.

The Methods now include the following statement:

“Quantification of NMT2/NAC co-sedimentation assays was performed in triplicate. Band intensities from immunoblots were measured in ImageJ, with NMT2 intensities weighted by RPL10a loading per lane.” Line 478

Additionally, immunoblots of the input samples for all co-sedimentation assays have been added to the supplementary data.

3. The part where the authors claim: "The N-terminal tail of NMT2 contains an unstructured region that has previously been shown necessary for ribosomal localization.²⁵ To investigate whether this region is also involved in NAC binding and subsequent recruitment to the ribosome, we generated an NMT2 Δ N mutant lacking the first 105 amino acids." raises important questions. **How does the NMT2 Δ N mutant test for NAC binding specifically rather than for general association with the complex (i.e. via other interactions)?** The claim of subsequent recruitment is also not supported by data and deserves a separate point.

The NMT2 Δ N mutant in the co-sedimentation assay shows whether the N-terminal tail interacts with NAC or the ribosome. If the tail was needed for NAC binding, we would see no increase in NMT2 Δ N localization when NAC is added. Since we still see increased localization with NAC present, this suggests the N-terminal tail likely interacts with the ribosome rather than directly with NAC.

We have now repeated the experiments and expanded this figure (now figure 6) to clearly show these differences.

4. The authors claim to test that "this region is also involved in NAC binding and subsequent recruitment to the ribosome" and that "First, the C-terminal tail of NAC β recruits NMT2 to the ribosome. NAC then orients NMT2 on the ribosome, with the NAC-NMT2 loop-barrel contact forming to stabilize the ternary complex." While presenting no data proving such sequential order of binding events. The authors need to either clearly present such data or tone down their statements to avoid this discrepancy.

This point has been addressed in our response to Reviewer 2, Comment 1, which provides key literature outlining the events leading to NAC binding to the ribosome. These references are now cited in the Discussion.

However, we also acknowledge the uncertainty regarding the precise order of NAC tail binding and orientation. Accordingly, we have toned down the related claims in both the Discussion section and the schematic in Figure 7.

5. The authors discuss that "NAC and NMT2 extends the polypeptide exit tunnel, generating a protected channel to the catalytic core of NMT2" and that "the protective channel formed by the ternary complex may be beneficial during co-translational nascent chain processing to shield the polypeptide". At the same time, much longer nascent chains were associated with NMT2 with same apparent efficiency (Fig 1E, F) and in the structure this "channel" seems to be at least half open (as presented in the scheme of Figure 6). The authors should reconcile this with their idea of the protective channel. In this sense, Figure 3 is quite misleading. Is there still room for the longer nascent chains to loop out of the tunnel as the structure and biochemical data suggest? Can the authors show that these interactions with longer nascent chains are still productive? It would greatly benefit the study to show the myristoylation activity directly for the reconstitution approach used for the structural part. Methods utilizing antibodies against myristoyl-glycine seem to be particularly useful here.

As described in our response to Reviewer 1 (Point 2), we tested the hypothesis that the NAC–NMT2 interface forms a protective channel by performing TEV protease digestion assays on stalled RNCs containing a TEV site within the nascent chain. While we observed modest protection in the presence of NMT2, the effect was not strong enough to support a definitive protective role. As a result, we revised the manuscript to tone down the interpretation of the channel's shielding function, including adjustments to the text and Figure 3. In parallel, we attempted to directly monitor NMT2-mediated myristoylation using two different pan–myristoyl-glycine antibodies, as suggested. Unfortunately, neither antibody produced interpretable results in our system.

6. The molecular model has several issues that have to be resolved: Chains s, t, 0 and RR are not supported by experimental data (density) and should not be included in the model. Model quality is generally poor in some aspects like the clashscore and especially the sidechain outliers, which would put this structure in the worst 6.4% in this regard. This is quite striking since the authors claim that "Any clashes were corrected using secondary structure, sidechain rotamer, nucleotide, and Ramachandran restraints." In the Methods.

We thank the reviewer for pointing this out. Chains s, t, RR, and 0 (see above figure- highlighted in red) correspond to ribosomal proteins located on the periphery of the small and large subunits, which are generally less resolved than proteins near the core of the ribosome. However, when the map is filtered to 6 Å and displayed at a contour level of 0.06, density corresponding to these proteins becomes visible. This is a common observation in cryo-EM, particularly for ribosomes, where peripheral proteins often exhibit increased mobility and are therefore less resolved at the reported average resolution.

In response to the reviewer's concern regarding model refinements, we have reviewed our model and then performed additional round of refinement in PHENIX, during which several sidechain outliers were corrected. This led to an overall improvement in the model's validation metrics reported in the cryo-EM validation table. We have revised the wording in the Methods section to more accurately reflect the remaining limitations of the model. The sentence now reads:

“The majority of clashes were corrected using secondary structure, sidechain rotamer, nucleotide, and Ramachandran restraints” Line 569

Reviewer #3- Minor points:

1. Both the NMT2 and NAC seem to be enriched in the 40S fraction in Fig 1B. How do the authors explain this? The authors should also include fractions 1 and 2 (which are not shown) so that the gradient profile is aligned with the WB fractions and the unbound fractions can be seen.

This point was addressed in our response to Reviewer 2, Minor Point 1. In our fractionation system, fraction 1 contains cell debris and membrane components and does not resolve well by SDS-PAGE. Nonetheless, to provide a complete profile, we repeated this experiment and now included fraction 2 in the updated gel shown in the revised Figure 1B. This addition improves alignment between the gradient profile and Western blot fractions and allows visualization of unbound proteins which trail into the 40S fractions.

2. The proposed histidine stacking interaction presented in Figure 5A is not clearly visible and was not observed in the model the authors have provided. For any claims on the importance of these residues in the interaction, the authors should present more convincing mutation analysis (e.g., mutating both H411 and H412 at the same time).

We attempted to generate a double mutant (H411A/H412A), but the protein could not be successfully purified, suggesting that this mutation may destabilize the protein or impair its folding. We also acknowledge that the resolution in this region is limited, which prevents confident modeling of a stacking interaction between the two histidine residues. However, similar histidine–histidine interactions have been reported in the literature.

To address this comment, we have revised the manuscript text to tone down the claim and now describe the residues as being “*in close proximity*” rather than explicitly forming a stacking interaction.

3. The authors write about “the ternary complex” in the Abstract, which has not been introduced. It would be better to specify the complex, especially since the term ternary complex has a very prominent established meaning in the translation field.

To avoid confusion, we have revised the abstract to explicitly name the components of the complex. The revised sentence now reads:

“Furthermore, the ribosome:NMT2:NAC complex is stabilized by a ribosomal RNA clamp that, together with NAC, orients NMT2 on the ribosomal surface for co-translational myristoylation of nascent chains.” Line 23

4. Figures 1A, 3A, Extended Data Figures 1, 6A-B and E, are not mentioned in the manuscript and sometimes latter panels are referred to before the initial ones (panel 4B for instance). The authors should fix these issues.

Thank you for the feedback. All citations have now been properly added.

5. The authors should specify and show how they “engineered a 3C cleavage site into the sequence” of NMT2. Was it necessary to mutate the second NMT2 residue to Pro to achieve the optimal the 3C protease cleavage motif?

The engineered 3C cleavage site in our construct is a canonical LEVLFQ↓ sequence, followed by “GA” which overlaps with the natural N-terminus of the MARCKS substrate. This design avoids the addition of extra residues and ensures that glycine is exposed at the N-terminus after 3C cleavage. The second residue of MARCKS was not mutated to proline. While the LEVLFQ/GP motif is optimal for 3C protease, previous studies have shown that the enzyme is also active on LEVLFQ/GA. This also demonstrated in our study in Extended Data Fig. 1A, which shows the removal of the N-terminal FLAG tag from the MARCKS constructs by 3C protease. A detailed sequence of the engineered region is shown below.

MDYKDHDGDKDHDIDYKDDDDKSSGLEVLFFQGAQFSKTAAKGE
 AAAERPGEAAVASSPSKANPVPYQPPFLCQWGRHQCAWKPLM

6. There is no "partial density" apparent for the myristoyl group in Extended Data Figure 6D. The authors should either clearly present such density or tone down their statement.

We agree with the reviewer the density for myristoyl group has been removed from the final deposited model and this statement has been toned down in the text and it now read as follows:

“Although the glycine residue is resolved within the catalytic site, no density for the myristoyl group (14-C) is visible, suggesting it may become flexible after being transferred to the nascent chain (Extended Data Fig 6D).” Line 156

7. There are no data presented for the statement "This anchor helix contains several positively charged residues that establish electrostatic interactions with uL23."

We thank the reviewer for pointing this out. We have now added references to Figures 4C and 4F to support this statement. Figure 4C illustrates the electrostatic interactions between the positively charged residues in the anchor helix and uL23. Figure 4F shows the full sequence alignment, now annotated to indicate the location of anchor helices D and E, highlighting the positively charged residues discussed in the text.

8. There seems to be no reaction for the "added to the reaction and pipetted gently" on line 394.

The wording of the sentence has been changed to “40μL of 2.8M NaOAc and 300μL of 95% EtOH were added to the resuspended pellet and pipetted gently to mix and left on ice for 5 minutes.” to more accurately describe the protocol. (Line 415)

9. There is nothing regarding the in vitro translation described above for the claim "crude RNCs were generated by reacting mRNA generated by in vitro transcription with RRLs (Promega, L4540) as described above". Even later, for the cryo-EM sample preparation, this should be described more clearly for non-expert readers.

An additional sentence describing the specifics of the in vitro translation reaction for crude RNCs has been added to the methods section NMT2 co-sedimentation with varying lengths of RNCMARCKS nascent chain constructs as follows:

“For the in vitro translation reaction to generate the stalled RNCs, 100μg of RNA was added to 200μL of RRL and supplemented with 0.04U/μL RNase Inhibitor (ThermoFisher REF EO0381),

0.5X Protease Inhibitor (Promega REF G6521), 24μM amino acid mix, 81μM KCl, and 2mM Mg(OAc)₂. The reaction was incubated for 25 minutes at 32°C, then placed on ice.” Line 423

To further clarify this section, we added the following:

“After the 25-minute incubation at 32°C and transfer to ice, the previously prepared in vitro translation reaction was loaded onto the column, which was sealed with parafilm and incubated for 2 hours at 4°C with constant rotation.” Line 503

10. Why was "pipetting up and down gently to avoid bubbles" used with the SDS sample buffer to prepare SDS PAGE samples? (Methods, line 436)

SDS can create bubbles when mixed too roughly. We pipetted gently to avoid bubbles during pellet resuspension and ensure accurate gel loading. This has been clarified in the Methods section.

11. What does it mean that Myristoyl-Coenzyme A "was spiked in"? (Methods, line 494)

We have revised the sentence to:

“Approximately 2 minutes prior to applying the sample to the grid, Myristoyl-Coenzyme A (Cayman Chemicals, #39777) was added to the sample to a final concentration of 100 μM.” Line 527

12. The Coenzyme A depicted in Fig 2C is presented as stick model, not cartoon as specified in the legend.

This has been updated.

13. Quantification of band intensities should be added to support the claims regarding data in Figure 1F and Figure 5B.

We have added quantifications for the gel in figure 1F and all the gels for Figure 5B.

14. Lines 202-203 - Sentence about the sequence alignment of the NMTs in the loop-barrel interaction region with the corresponding link to Figure 5C is missing and this panel is only referred to at the end of Discussion.

The following sentence has been added to clarify the link to the sequence alignment (now Figure panel 5B)

“A sequence alignment demonstrates these residues are highly conserved across most eukaryotes.”

15. Lines 282-283 - It would be fair to mention that the observed interaction between NMT2 and NACβ, that is hypothesized to stabilize NMT2 in favorable conformation relative to

the nascent chain, is not universally conserved, as the *A. thaliana* NMT2 is clearly missing both histidine residues, as depicted in the alignment in Figure 5C.

The sentence added to address the previous point (line 206) addresses this as well, directly stating that the residues are highly conserved across *most* eukaryotes to account for the lack of universal conservation.

16. The authors should check and correct the units they are using in the Methods section (RPM instead of g throughout Methods, concentration of A260, $\epsilon/\text{\AA}^2$).

Units have been corrected.

17. Lines 402, 420, 435 - Rotor used for the ultracentrifugation is not specified.

Methods have been updated at the listed locations with information about the rotors used.

Line 427- “*The reactions were then spun at 20,821 $\times g$ at 4°C for 10 minutes in a Microliter 30 x 2mL fixed-angle rotor.*”

Line 446- “*The reactions were centrifuged at 355,040 $\times g$ for 1hr at 4°C using a TLA 120.1 rotor.*”

Line 460- “*The remaining reactions were centrifuged at 355,040 $\times g$ for 1hr at 4°C using a TLA 120.1 rotor.*”

18. Software versions used in the study should be specified in the Methods.

Software listed in the Methods has had the versions added as follows:

ChimeraX (v1.6.1). Lines 565, 571

Coot (v0.9.8.93) Line 566

PHENIX (v1.20.1). Line 569

CryoSPARC (V4.6.0). Lines 539, 560, 909

19. Line 684 - Year of the cited article is missing.

This citation has been updated to add year of article publication— Shanmuganathan V, Schiller N, Magoulopoulou A, et al. Structural and mutational analysis of the ribosome-arresting human XBP1u. Hegde RS, Ron D, Mankin A, eds. *eLife*. 2019;8:e46267. doi:10.7554/eLife.46267

Dear Prof. Jomaa,

Thank you for submitting a revised version of your manuscript. Your study has now been seen the original referee #3, who finds that the previous concerns have been addressed and now recommends publication of the manuscript. There remain only a few mainly editorial points that have to be addressed before I can extend formal acceptance of the manuscript:

- On the abstract page of the manuscript, please include 4-5 general keyword terms to enhance searchability.
- Please adjust the format of the reference list and of the in-text citations according to EMBO Journal format (alphabetical order, author name et al + year.../up to 10 author names in the reference list before et al / please refer to our Guide to Authors for additional information on EMBO J reference format).
- Please rename the "Data and materials availability" section to "Data Availability"
- Please rename the Conflict of Interest section into "Disclosure and Competing Interests Statement", in accordance with our updated Guide to Authors (<https://www.embopress.org/competing-interests>)
- As we are switching from a free-text author contribution statement towards a more formal statement based on Contributor Role Taxonomy (CRediT) terms, please remove the present Author Contribution section and instead specify each author's contribution(s) directly in the Author Information page of our submission system during upload of the final manuscript. See <https://casrai.org/credit/> for more information.
- Please adjust the in-text callouts for individual figures and figure panels: e.g. all callouts should be listed sequentially
- Please rename the expanded view figures to Figure EV1-EV6 instead of Figure Expanded View 1-6 in Figure legends
- Please make sure that the title page of the appendix file starts with "Appendix for + ms title" and contains ToC with the page numbers for the listed items
- Please provide suggestions for a short 'blurb' text prefacing and summing up the conceptual aspect of the study in two sentences (max. 250 characters), followed by 3-5 one-sentence 'bullet points' with brief factual statements of key results of the paper; they will form the basis of an editor-written 'Synopsis' accompanying the online version of the article. Please also provide an altered synopsis image, making sure that the aspect ratio conforms to our website's format - it should be exactly 550 pixels wide and between 300-600 pixels high.
- The blots in Figures 5C input and Figure 6C are heavily over-exposed, also in the source data. Please check the original captured image and remake these figures, updating the source data.
- Please add the specific URLs for EMD-49275, PDB 9NDP datasets in the data availability statement.
- Please adjust the order of the manuscript sections: Title page with complete author information, Abstract, Keywords, Introduction, Results, Discussion, Methods, Data Availability Section, Acknowledgements, Disclosure and Competing Interests Statement, References, Main figure legends, Tables, Expanded Figure Legends.
- Figure Legends (main + EV):
 1. Please note that the exact p values are not provided in the legends of figures 6E, F.

With best regards,

Cornelius Schneider

Cornelius Schneider, PhD
Editor | The EMBO Journal
c.schneider@embojournal.org

Please discuss the revision progress ahead of this time with the editor if you require more time to complete the revisions. Use the link below to submit your revision:

Referee #3:

The authors have addressed all major and minor revision points I have raised. I support the revised manuscript for publication.

Point by point answers to editorial comments:

On the abstract page of the manuscript, please include 4-5 general keyword terms to enhance searchability.

Keywords have been added to the bottom of the Abstract page as follows:

Keywords: NMT2, myristoylation, ribosome, myristoyltransferase, cryo-EM, co-translational processing

Please adjust the format of the reference list and of the in-text citations according to EMBO Journal format (alphabetical order, author name et al + year.../up to 10 author names in the reference list before et al / please refer to our Guide to Authors for additional information on EMBO J reference format).

In text citations and reference list have been updated to follow EMBO Journal format.

Please rename the "Data and materials availability" section to "Data Availability"

This section has been renamed.

Please rename the Conflict of Interest section into "Disclosure and Competing Interests Statement", in accordance with our updated Guide to Authors (<https://www.embopress.org/competing-interests>)

This section has been renamed.

As we are switching from a free-text author contribution statement towards a more formal statement based on Contributor Role Taxonomy (CRediT) terms, please remove the present Author Contribution section and instead specify each author's contribution(s) directly in the Author Information page of our submission system during upload of the final manuscript. See <https://casrai.org/credit/> for more information.

The author contribution section has been removed from the manuscript and author contributions are described in the manuscript submission system.

Please adjust the in-text callouts for individual figures and figure panels: e.g. all callouts should be listed sequentially

Figure callouts have been updated to be sequentially appearing in the text.

Please rename the expanded view figures to Figure EV1-EV6 instead of Figure Expanded View 1-6 in Figure legends

These figures have been renamed.

Please make sure that the title page of the appendix file starts with "Appendix for + ms title" and contains ToC with the page numbers for the listed items

A title page and table of contents has been added to the Appendix file.

Please provide suggestions for a short 'blurb' text prefacing and summing up the conceptual aspect of the study in two sentences (max. 250 characters), followed by 3-5 one-sentence 'bullet points' with brief factual statements of key results of the paper; they will form the basis of an editor-written 'Synopsis' accompanying the online version of the article. Please also provide an altered synopsis image, making sure that the aspect ratio conforms to our website's format - it should be exactly 550 pixels wide and between 300-600 pixels high.

A synopsis blurb and synopsis image have been uploaded with the resubmission.

The blots in Figures 5C input and Figure 6C are heavily over-exposed, also in the source data. Please check the original captured image and remake these figures, updating the source data.

The blot exposure has been adjusted in the figures as well as the source data images.

Please add the specific URLs for EMD-49275, PDB 9NDP datasets in the data availability statement.

The URLs (listed below) for the EMDB and PDB entries have been added to the text.

EMDB: <https://www.ebi.ac.uk/emdb/EMD-49275>

PDB: <https://www.rcsb.org/structure/9NDP>

Please adjust the order of the manuscript sections: Title page with complete author information, Abstract, Keywords, Introduction, Results, Discussion, Methods, Data Availability Section, Acknowledgements, Disclosure and Competing Interests Statement, References, Main figure

legends, Tables, Expanded Figure Legends.

Manuscript section order has been adjusted to follow what is listed above.

Figure Legends (main + EV):

1. Please note that the exact p values are not provided in the legends of figures 6E, F.

P-values for Figures 6E and 6F are listed in the legends (highlighted below).

E) A quantification of the co-sedimentation results from the co-sedimentation shown in C.

Assays were completed as 3 independent replicates and band intensity was weighted by RPL10a band intensity per lane and normalized to WT NMT2 and significance was determined using a one-way ANOVA, and uncorrected Fisher's LSD post-hoc test. P-values are 0.0038 (WT NAC),

0.0079 (NAC α Δ C), 0.0068 (NAC α Δ N), and 0.2936 (NAC β Δ C), respectively. **F)** A

quantification of the co-sedimentation results from the co-sedimentation shown in D. Assays were completed as 3 independent replicates and band intensity was weighted by RPL10a band intensity per lane and normalized to WT NMT2 and significance was determined using a two-way ANOVA, and Tukey's multiple comparisons post-hoc test. All pairwise comparisons were significant besides WT NMT2 alone vs. WT NMT2 + NAC β Δ C (p=0.9930), and NMT2 Δ N alone vs. NMT2 Δ N + NAC β Δ C (p=0.2122). P-values for all other comparisons are <0.0001, except for NMT2 Δ N alone vs. NMT2 Δ N + WT NAC (p=0.0002).

Two p-values are below 0.0001, and PRISM does not provide exact p-values for below this threshold. The PRISM file is provided with the source data during resubmission and can be referenced there.

Dear Prof. Jomaa,

I am pleased to inform you that your manuscript has been accepted for publication in the EMBO Journal.

Yours sincerely,

Cornelius Schneider, PhD
Editor
The EMBO Journal
c.schneider@embojournal.org
